# Distributional Signals for Node Classification in Graph Neural Networks

## Abstract

In graph neural networks (GNNs), both node features and labels are examples of graph signals, a key notion in graph signal processing (GSP). While it is common in GSP to impose signal smoothness constraints in learning and estimation tasks, it is unclear how this can be done for discrete node labels. We bridge this gap by introducing the concept of distributional graph signals. In our framework, we work with the distributions of node labels instead of their values and propose notions of smoothness and non-uniformity of such distributional graph signals. We then propose a general regularization method for GNNs that allows us to encode distributional smoothness and non-uniformity of the model output in semi-supervised node classification tasks. Numerical experiments demonstrate that our method can significantly improve the performance of most base GNN models in different problem settings.

## 1 Introduction

We consider the semi-supervised node classification problem (Kipf & Welling, 2017) that determines class labels of nodes in graphs given sample observations and possibly node features. Numerous graph neural network (GNN) models have been proposed to tackle this problem. One of the first models is the graph convolutional network (GCN) (Defferrard et al., 2016). Interpreted geometrically, a GCN aggregates information such as node features from the neighborhood of each node of the graph. Algebraically, this process is equivalent to applying a graph convolution filter to node feature vectors. Subsequently, many GNN models with different considerations are introduced. Popular models include the graph attention network (GAT) (Veličković et al., 2018) that learns weights between pairs of nodes during aggregation, and the hyperbolic graph convolutional neural network (HGCN) (Chami et al., 2019) that considers embedding of nodes of a graph in a hyperbolic space instead of a Euclidean space. For inductive learning, GraphSAGE (Hamilton et al., 2017) is proposed to generate low-dimensional vector representations for nodes that are useful for graphs with rich node attribute information. While new models draw inspiration from GCN, GCN itself is built upon the foundation of graph signal processing (GSP).

GSP is a signal processing framework that handles graph-structured data (Shuman et al., 2013; Ortega et al., 2018; Ji & Tay, 2019). A graph signal is a vector with each component corresponding to a node of a graph. Examples include node features and node labels. Moreover, convolutions used in models such as GCN are special cases of convolution filters in GSP (Shuman et al., 2013). All these show the close connections between GSP theory and GNNs.

In GSP, signal smoothness (over the graph) is widely used to regularize inference tasks. Intuitively, a signal is smooth if its values are similar at each pair of nodes connected by an edge. One popular way to formally define signal smoothness is to use the Laplacian quadratic form. There are numerous GSP tools that leverage a smooth prior of the graph signals. For example, Laplacian (Tikhonov) regularization is proposed for noise removal in Shuman et al. (2013) and signal interpolation (Narang et al., 2013). In Chen et al. (2015), it is used in graph signal in-painting and anomaly detection. In Kalofolias (2016), the same technique is used for graph topology inference.

However, for GNNs, it is remarked in Yang et al. (2021, Section 4.1.2) that "graph Laplacian regularization can hardly provide extra information that existing GNNs cannot capture". Therefore, a regularization scheme based on feature propagation is proposed. It is demonstrated to be effective by comparing with other methods such as Feng et al. (2021) and Deng & Zhu (2019) based on

adversarial learning and Stretcu et al. (2019) that co-trains GNN models with an additional agreement model, which gives the probability that two nodes have the same label. We partially agree with the above assertion regarding graph Laplacian regularization, while remaining reservative about its full correctness. In this paper, we propose a method that is inspired by Laplacian regularization. As our main contribution, we introduce the notion of distributional graph signals, instead of considering graph signals. Analogous to the graph signal smoothness defined using graph Laplacian, we define the smoothness of distributional graph signals. Together with another property known as non-uniformity, we devise a regularization scheme for GNNs in node classification tasks. This approach is easy to implement and can be used as a plug-in regularization term together with any given base GNN model. Its effectiveness is demonstrated with numerical results.

## 2 DISTRIBUTIONAL GRAPH SIGNALS

In this section, we motivate and introduce distributional graph signals based on GSP theory.

### 2.1 GSP PRELIMINARIES AND SIGNAL SMOOTHNESS

In this subsection, we give a brief overview of GSP theory (Shuman et al., 2013). The focus is on the discussion of graph signal smoothness.

Let $\mathcal{G} = (\mathcal{V}, \mathcal{E})$ be an undirected graph with $\mathcal{V}$ the vertex set and $\mathcal{E}$ the edge set. Suppose the size of the graph is $n = |\mathcal{V}|$. Fix an ordering of $\mathcal{V}$. Then, the *space of graph signals* can be identified with the vector space $\mathbb{R}^n$, with a graph signal $\boldsymbol{x} \in \mathbb{R}^n$, which assigns its $i$-th component to the $i$-th vertex of $\mathcal{G}$. By convention, signals are in column form, and $\boldsymbol{x}(i)$ is the $i$-th component of $\boldsymbol{x}$.

In GSP, the key notion is *the graph shift operator*. Though there are several choices for the graph shift operator, in our paper, we consider a common choice: $\boldsymbol{L}_{\mathcal{G}}$, the Laplacian of $\mathcal{G}$, defined by $\boldsymbol{L}_{\mathcal{G}} = \boldsymbol{D}_{\mathcal{G}} - \boldsymbol{A}_{\mathcal{G}}$, where $\boldsymbol{D}_{\mathcal{G}}, \boldsymbol{A}_{\mathcal{G}}$ are the degree matrix and adjacency matrix of $\mathcal{G}$, respectively. The Laplacian $\boldsymbol{L}_{\mathcal{G}}$ is positive semi-definite and symmetric. By the spectral theorem, it has an eigendecomposition $\boldsymbol{L}_{\mathcal{G}} = \boldsymbol{U}_{\mathcal{G}} \Lambda_{\mathcal{G}} \boldsymbol{U}_{\mathcal{G}}^{\top}$. In the decomposition, $\Lambda_{\mathcal{G}}$ is a diagonal matrix, whose diagonal entries $\{\lambda_1, \dots, \lambda_n\}$ are eigenvalues of $\boldsymbol{L}_{\mathcal{G}}$. They are non-negative and we assume $\lambda_1 \leq \dots \leq \lambda_n$. The associated eigenbasis $\{\boldsymbol{u}_1, \dots, \boldsymbol{u}_n\}$ are the columns of $\boldsymbol{U}_{\mathcal{G}}$. In GSP, an eigenvector with a small eigenvalue (and hence a small index) is considered to be smooth. The signal values of such a vector have small fluctuations across the edges of $\mathcal{G}$.

Given a graph signal $\boldsymbol{x}$, its *graph Fourier transform* is $\hat{\boldsymbol{x}} = \boldsymbol{U}_{\mathcal{G}}^{\top} \boldsymbol{x}$, or equivalently, $\hat{\boldsymbol{x}}(i) = \langle \boldsymbol{x}, \boldsymbol{u}_i \rangle$, for $1 \leq i \leq n$. The components $\hat{\boldsymbol{x}}(i)$ of $\hat{\boldsymbol{x}}$ are called the *frequency components* of $\boldsymbol{x}$. Same as above, the signal $\boldsymbol{x}$ is smooth if $\hat{\boldsymbol{x}}(i)$ has a small absolute value for large $i$. Quantitatively, we can define its *total variation* by

$$\mathcal{T}(\boldsymbol{x}) = \sum_{(v_i, v_j) \in \mathcal{E}} (\boldsymbol{x}(i) - \boldsymbol{x}(j))^2 = \boldsymbol{x}^{\top} \boldsymbol{L}_{\mathcal{G}} \boldsymbol{x}. \tag{1}$$

It is straightforward to compute that $\mathcal{T}(\boldsymbol{u}_i) = \lambda_i$. This observation indicates that it is reasonable to use total variation as a measure of smoothness. Minimizing the total variation of graph signals has many applications in GSP as we have pointed out in Section 1.

### 2.2 STEP GRAPH SIGNALS

Let $\mathbb{S}$ be a finite set of numbers. A *step graph signal with respect to (w.r.t.)* $\mathbb{S}$ is a graph signal $\boldsymbol{x}$ such that all its components take values in $\mathbb{S}$, i.e., $\boldsymbol{x} \in \mathbb{S}^n$.

**Example 1.** *For the simplest example, consider the classical Heaviside function $H$ on $\mathbb{R}$ defined by $H(x) = 1$, for $x > 0$ and $H(x) = 0$, for $x \leq 0$. It is a non-smooth function, as it is not even continuous at $x = 0$. On the other hand, let $\mathcal{G}$ be the path graph with $2m + 1$ nodes embedded on the real line by identifying the nodes of $\mathcal{G}$ with the integers in the interval $[-m, m]$. Then $H$ induces a step graph signal $\boldsymbol{h}$ on $\mathcal{G}$. Same as the Heaviside function $H$, the signal $\boldsymbol{h}$ should be considered to be a non-smooth graph signal.*

Step graph signals occur naturally in semi-supervised node classification tasks. In particular, if $\mathbb{S}$ is the set of all possible class labels, then the labels of the nodes of $\mathcal{G}$ form a step graph signal $\boldsymbol{c}$ w.r.t. $\mathbb{S}$

on $\mathcal{G}$. We expect that analogous to Example 1, $\boldsymbol{c}$ can possibly be non-smooth. To demonstrate, we analyze $\boldsymbol{c}$ using its Fourier transform $\hat{\boldsymbol{c}}$ (cf. Section 2.1). More specifically, we take $\mathcal{G}_0$ to be the main connected component of the Cora graph Sen et al. (2008) and $\boldsymbol{c}$ to be the ground truth labels. We also generate a random signal $\boldsymbol{r}$ following the same empirical distribution (on $\mathbb{S}$) estimated using $\boldsymbol{c}$. We show plots of both Fourier transforms $\hat{\boldsymbol{c}}$ and $\hat{\boldsymbol{r}}$ in Fig. 1. We see that the high-frequency components of the ground truth labels $\boldsymbol{c}$ can be large, and it is even possible that its spectrum, i.e., frequency components, resemble that of a random signal. The observations support our speculation about the non-smoothness of the step signals. Therefore, in order to leverage signal smoothness to enhance model performance, we need to find an alternative to the step (label) signals. This also supports the remark of Yang et al. (2021) regarding Laplacian regularization from a different point of view (see also the experiments in Section 4.2).

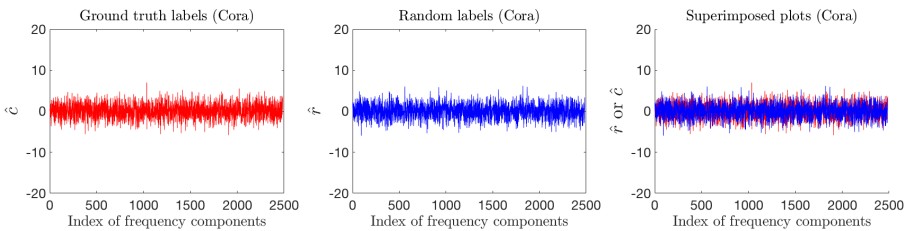

Figure 1: Plots of $\hat{\boldsymbol{c}}$ and $\hat{\boldsymbol{r}}$ for Cora.

## 2.3 DISTRIBUTIONAL GRAPH SIGNALS AND MARGINALS

We want to use probability theory to introduce the notion of "smoothness" for step signals in the next section. For preparation, in this subsection, we formally introduce distributional graph signals and discuss how they arise naturally in GNNs. Our theory (e.g., in Definition 1 and equation (3)) is based on probability measures defined on metric spaces. To this end, we endow any discrete space $\mathbb{S}$ with the discrete metric $d(s_1, s_2) = 1$ if $s_1 \neq s_2 \in \mathbb{S}$ and 0 otherwise. For $\mathbb{R}^n$, one can use the usual Euclidean metric or any other norm.

**Definition 1.** *For a metric space* $\mathbb{M} = \mathbb{S}^n$ *or* $\mathbb{R}^n$, *let* $\mathcal{P}(\mathbb{M})$ *be the space of probability measures on* $\mathbb{M}$ *(w.r.t. the Borel $\sigma$-algebra) having finite second moments. An element* $\mu \in \mathcal{P}(\mathbb{M})$ *is called a* distributional graph signal. *The marginals of a distributional graph signal* $\mu$ *are the marginal distributions* $\mathcal{N} = \{\mu_i : 1 \leq i \leq n\}$ *w.r.t. the $n$ coordinates of either* $\mathbb{S}^n$ *or* $\mathbb{R}^n$.

To understand why a distribution $\mu$ is related to GSP, consider a step graph signal (resp. ordinary graph signal) $\boldsymbol{x}$ in $\mathbb{S}^n$ (resp. $\mathbb{R}^n$). It induces the delta distribution $\delta_{\boldsymbol{x}}$ in $\mathcal{P}(\mathbb{S}^n)$ supported at $\boldsymbol{x}$. Its marginals are the delta distributions $\{\delta_{\boldsymbol{x}(i)} : 1 \leq i \leq n\}$. Therefore, Definition 1 subsumes ordinary graph signals as special cases.

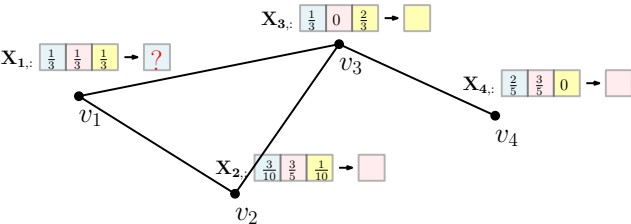

Figure 2: We give an example of the marginals of a distributional graph signal. We notice that for $\boldsymbol{X}_{1,:}$, the probability weights are equal and it is hard to determine the $\arg \max$ of the 3 classes. Moreover, the weights of the 2nd class have large differences along a few edges, e.g., $(v_2, v_3), (v_3, v_4)$. These features make prediction unreliable. We propose solutions to such issues in this paper.

Distributional graph signals also occur naturally in GNN models (illustrated in Fig. 2). As in the previous subsection, let $\mathbb{S}$ be the set of all possible class labels of size $m$ and $M_{n,m}(\mathbb{R})$ be the space of $n \times m$ matrices. For a typical model $\mathcal{M}$ such as the GCN, the last stage of the pipeline usually

consists of the following steps[1]:

$$\text{Logits: } \boldsymbol{O} \in M_{n,m}(\mathbb{R}) \xrightarrow{\text{Softmax:}\phi} \boldsymbol{X} = \phi(\boldsymbol{O}) \in M_{n,m}(\mathbb{R}) \xrightarrow{\text{arg max}} \text{Output labels: } \boldsymbol{c} \in \mathbb{S}^n. \quad (2)$$

The $i$-th row $\boldsymbol{X}_{i,:}$ of $\boldsymbol{X}$ can be viewed as the weights of a probability distribution $\mu_{\boldsymbol{X},i}$ on $\mathbb{S}$. They do not directly give a distributional graph signal in $\mathcal{P}(\mathbb{S}^n)$. However, we shall interpret $\mathscr{N}_{\boldsymbol{X}} = \{\mu_{\boldsymbol{X},i}, 1 \leq i \leq n\}$ as the marginals of some unknown distributional graph signal, and $\mathscr{N}_{\boldsymbol{X}}$ is the main subject of study in this paper. In order to mimic the ways smoothness of graph signals is used in GSP, we introduce an appropriate notion of the total variation of distributional graph signals in the next section.

## 3 REGULARIZED GRAPH NEURAL NETWORKS

In this section, we study smoothness and non-uniformity properties of distributional graph signals in Section 3.1 and Section 3.2 respectively. Each of these subsections yields an expression that we want to minimize. They are combined in Section 3.3 to give the proposed regularization term.

### 3.1 TOTAL VARIATIONS AND LAPLACIAN REGULARIZATION

The goal of this subsection is to introduce and compare different notions of total variation associated with distributional graph signals and their marginals that leverage signal smoothness. First of all, given $\mu \in \mathcal{P}(\mathbb{M})$ for $\mathbb{M} = \mathbb{S}^n$ or $\mathbb{R}^n$, its total variation can be modified directly from (1) as follows:

$$\mathcal{T}(\mu) = \mathbb{E}_{\boldsymbol{x} \sim \mu} \mathcal{T}(\boldsymbol{x}) = \int \sum_{(v_i, v_j) \in \mathcal{E}} d(\boldsymbol{x}(i), \boldsymbol{x}(j))^2 \, \mathrm{d}\mu(\boldsymbol{x}). \quad (3)$$

where $d$ is the metric on $\mathbb{S}$ or $\mathbb{R}$.

However, in many cases of interest such as GNNs, only marginals of some $\mu$ are observed (cf. Section 2.3). We want to define total variation in such a situation as well. We borrow ideas from the prototype of the Wasserstein metric (Villani, 2009), which we recall now. The notations and assumptions follow those of Definition 1.

**Definition 2.** *For $\mu_1, \mu_2 \in \mathcal{P}(\mathbb{M})$, the* Wasserstein metric $W(\mu_1, \mu_2)$ *between $\mu_1, \mu_2$ is defined by*

$$W(\mu_1, \mu_2)^2 = \inf_{\gamma \in \Gamma(\mu_1, \mu_2)} \int d(x,y)^2 \, \mathrm{d}\gamma(x,y),$$

*where $\Gamma(\mu_1, \mu_2)$ is the set of* couplings *of $\mu_1, \mu_2$, i.e., the collection of probability measures on $\mathbb{M} \times \mathbb{M}$ whose marginals are $\mu_1$ and $\mu_2$, respectively.*

It can be verified that $W(\cdot, \cdot)$ indeed defines a metric on $\mathcal{P}(\mathbb{M})$ (see Villani (2009)). As a special case, if $\delta_x$ and $\delta_y$ are delta distributions supported on $x, y \in \mathbb{M}$, then $W(\delta_x, \delta_y) = d(x,y)$. The key insight is that we want to take infimum over all possible distributions given the prescribed marginals; and whatever we define, it should subsume (1) as a special case for a collection of delta distributions.

**Definition 3.** *Given $\mathscr{N} = \{\mu_i : 1 \leq i \leq n\}$ with $\mu_i \in \mathcal{P}(\mathbb{S})$ (resp. $\mu_i \in \mathcal{P}(\mathbb{R})$), for $1 \leq i \leq n$, then the total variation of $\mathscr{N}$ is defined as:*

$$\mathcal{T}(\mathscr{N}) = \inf_{\mu \in \Gamma(\mathscr{N})} \mathcal{T}(\mu), \quad (4)$$

*where $\Gamma(\mathscr{N})$ is the collection of all distributional graph signals in $\mathcal{P}(\mathbb{S}^n)$ (resp. $\mathcal{P}(\mathbb{R}^n)$) whose marginals agree with $\mathscr{N}$.*

Though an important theoretical tool, the Wasserstein metric is usually difficult to compute explicitly. On the other hand, if $\mathcal{G}$ is the graph with 2 nodes connected by an edge and $\mathscr{N} = \{\mu_1, \mu_2\}$, then $\mathcal{T}(\mathscr{N}) = W(\mu_1, \mu_2)^2$. As a consequence, finding the exact value of $\mathcal{T}(\mathscr{N})$ can be challenging. Therefore, we next introduce approximations that can be more readily computed.

---

[1]To simplify the presentation, we assume that all models considered have the intermediate softmax step, though some implementations omit this part using the fact that the exponential function is increasing.

For $\mathscr{N} = \{\mu_i : 1 \leq i \leq n\}$ with $\mu_i \in \mathcal{P}(\mathbb{S})$, let $\boldsymbol{\mu}_s \in \mathbb{R}^n$ be the graph signal of probability weights of $s \in \mathbb{S}$, i.e., $\boldsymbol{\mu}_s(i)$ is the probability weight of $\mu_i$ at $s$. We can also stack these signals as a matrix $\boldsymbol{X}_{\mathscr{N}}$ whose columns are $\boldsymbol{\mu}_s$. Based on the GSP version of total variation (cf. (1)), we introduce two more versions of total variation that are easy to compute.

**Definition 4.** *Given $\mathscr{N} = \{\mu_i : 1 \leq i \leq n\}$ with $\mu_i \in \mathcal{P}(\mathbb{S})$, we define the $\ell^1$ and $\ell^2$ versions of total variations as:*

- $\mathcal{T}_1(\mathscr{N}) = \sum_{s \in \mathbb{S}} \sum_{(v_i, v_j) \in \mathcal{E}} |\boldsymbol{\mu}_s(i) - \boldsymbol{\mu}_s(j)|$, *and*

- $\mathcal{T}_2(\mathscr{N}) = \sum_{s \in \mathbb{S}} \sum_{(v_i, v_j) \in \mathcal{E}} \left(\boldsymbol{\mu}_s(i) - \boldsymbol{\mu}_s(j)\right)^2 = \mathrm{Tr}(\boldsymbol{X}_{\mathscr{N}}^\top \boldsymbol{L}_{\mathcal{G}} \boldsymbol{X}_{\mathscr{N}})$, *where* $\mathrm{Tr}$ *is the matrix trace.*

The complexity of computing either $\mathcal{T}_1$ or $\mathcal{T}_2$ is at most $\mathcal{O}(|\mathbb{S}||\mathcal{E}|)$, and it involves only matrix multiplication for $\mathcal{T}_2$. In addition, $\mathcal{T}, \mathcal{T}_1$ and $\mathcal{T}_2$ satisfy the following relation.

**Theorem 1.** *Given $\mathscr{N} = \{\mu_i : 1 \leq i \leq n\}$ with $\mu_i \in \mathcal{P}(\mathbb{S})$, we have*

$$\mathcal{T}_2(\mathscr{N}) \leq \mathcal{T}_1(\mathscr{N}) \leq 2\mathcal{T}(\mathscr{N}).$$

*Moreover, $\mathcal{T}_1(\mathscr{N}) = 2\mathcal{T}(\mathscr{N})$ if $\mathcal{G}$ is a tree.*

The discussion and proof of a more general result can be found in Appendix D. The upshot is that if we expect $\mathcal{T}(\mathscr{N})$ to be small for some $\mathscr{N}$, then so are necessarily $\mathcal{T}_1(\mathscr{N})$ and $\mathcal{T}_2(\mathscr{N})$. In view of computation cost, we mainly use $\mathcal{T}_2$ in the design of the regularization model in Section 3.3. Note that this is analogous to the Laplacian regularization in GSP.

## 3.2 Non-uniformity

As discussed in Section 2.3, a base GNN model may output a matrix $\boldsymbol{X}$ (in (2)), with associated $\mathscr{N}_{\boldsymbol{X}} = \{\mu_{\boldsymbol{X},i}, 1 \leq i \leq n\}$. For node classification problems, it is desirable that there is less ambiguity in the decision for each node so that one can pinpoint the correct class label. Mathematically, this requires that each $\mu_{\boldsymbol{X},i}$ deviates from the uniform distribution $\mathcal{U}(\mathbb{S})$ on the finite set of label classes $\mathbb{S}$, measured by the Wasserstein metric $W(\mu_{\boldsymbol{X},i}, \mathcal{U}(\mathbb{S}))$. The following result is proved in Appendix D.

**Lemma 1.** *For a fixed sequence of non-positive numbers $(a_i)_{1 \leq i \leq n}$, there is a constant $C$ independent of $\boldsymbol{X}$ such that*

$$\mathrm{Tr}(\boldsymbol{X}^\top \boldsymbol{D} \boldsymbol{X}) + C \geq 2 \sum_{1 \leq i \leq n} a_i W(\mu_{\boldsymbol{X},i}, \mathcal{U}(\mathbb{S}))^2, \tag{5}$$

*where $\boldsymbol{D}$ is the diagonal matrix with diagonal entries $(a_i)_{1 \leq i \leq n}$. Moreover, $\mathrm{Tr}(\boldsymbol{X}_o^\top \boldsymbol{D} \boldsymbol{X}_o) \leq \mathrm{Tr}(\boldsymbol{X}^\top \boldsymbol{D} \boldsymbol{X}) \leq \mathrm{Tr}(\boldsymbol{X}_u^\top \boldsymbol{D} \boldsymbol{X}_u)$, where $\boldsymbol{X}_o$ is a matrix with each row a one-hot vector and $\boldsymbol{X}_u$ is the matrix with each entry $1/|\mathbb{S}|$.*

As we want each $\mu_{\boldsymbol{X},i}$ to deviate from the uniform distribution, the right-hand side of (5) should be made small (as negative as possible). This is ensured if the proxy $\mathrm{Tr}(\boldsymbol{X}^\top \boldsymbol{D} \boldsymbol{X})$, which is easy to compute, is small. Moreover, we notice that $\boldsymbol{X}_u$ (resp. $\boldsymbol{X}_o$) corresponds to uniform (resp. $\delta$) marginal distributions. The second half of the statement suggests that minimizing $\mathrm{Tr}(\boldsymbol{X}^\top \boldsymbol{D} \boldsymbol{X})$ may drive marginals, i.e., rows of $\boldsymbol{X}$, to comply with non-uniformity. Numerical experiments in Section 4.2 support that this proxy works well. We use this term in conjunction with $\mathcal{T}_2$ introduced in Section 3.1 in our proposed regularized model.

## 3.3 The loss function with regularization

Suppose we are given a base GNN model, denoted by $\mathcal{M}$. Let $\boldsymbol{F}$ be the matrix of input feature vectors and $\Theta$ be the parameter space for $\mathcal{M}$. Assume $\mathcal{M}$ has a loss function $\mathcal{L}_{\mathcal{M}}$, and the model is set to solve the optimization problem $\min_{\theta \in \Theta} \mathcal{L}_{\mathcal{M}}(\boldsymbol{F}, \theta)$.

Consider the steps given in (2). We have $\boldsymbol{X} = \phi(\boldsymbol{O})$ viewed as a matrix of probability weights. For regularization, we introduce another loss $\mathcal{L}_0$ to supplement $\mathcal{L}_{\mathcal{M}}$. Sections 3.1 and 3.2 suggest that $\mathcal{L}_0$ should consist of two parts $\mathcal{L}_0 = \mathcal{L}_1 + \mathcal{L}_2$. The loss $\mathcal{L}_1$ (cf. Definition 4) is related to the smoothness

of the distributional graph signal $\boldsymbol{X}$ and takes the form $\mathcal{L}_1(\boldsymbol{X}) = \mathrm{Tr}(\boldsymbol{X}^\top \boldsymbol{L}_{\mathcal{G}} \boldsymbol{X}) = \mathrm{Tr}(\boldsymbol{X}^\top (\boldsymbol{D}_{\mathcal{G}} - \boldsymbol{A}_{\mathcal{G}})\boldsymbol{X})$. On the other hand, $\mathcal{L}_2$ prevents the distributional signal from being uniform and can be explicitly expressed as $\mathcal{L}_2(\boldsymbol{X}) = \mathrm{Tr}(\boldsymbol{X}^\top \boldsymbol{D} \boldsymbol{X})$ for a suitably chosen negative semi-definite diagonal matrix $\boldsymbol{D}$ (cf. Lemma 1). Summing up $\mathcal{L}_1$ and $\mathcal{L}_2$, we have $\mathcal{L}_0(\boldsymbol{X}) = \mathrm{Tr}\left(\boldsymbol{X}^\top (\boldsymbol{D}_{\mathcal{G}} - \boldsymbol{A}_{\mathcal{G}} + \boldsymbol{D})\boldsymbol{X}\right)$. In our experiments, we take $\boldsymbol{D} = \boldsymbol{I}_n - \boldsymbol{D}_{\mathcal{G}}$ and obtain the easily computable loss

$$\mathcal{L}_0(\boldsymbol{X}) = \mathrm{Tr}\left(\boldsymbol{X}^\top (\boldsymbol{I}_n - \boldsymbol{A}_{\mathcal{G}})\boldsymbol{X}\right).$$

In summary, if we express the output of the second last layer of $\mathcal{M}$ as $\boldsymbol{O} = \psi(\boldsymbol{F}, \theta)$ of input $\boldsymbol{F}$ and model parameters $\theta$, then the regularized model R-$\mathcal{M}$ of $\mathcal{M}$ solves the optimization:

$$\min_{\theta \in \Theta} \mathcal{L}_{\text{R-}\mathcal{M}}(\boldsymbol{F}, \theta) = \min_{\theta \in \Theta} \mathcal{L}_{\mathcal{M}}(\boldsymbol{F}, \theta) + \eta \cdot \mathcal{L}_0\left(\phi \circ \psi(\boldsymbol{F}, \theta)\right), \tag{6}$$

where the coefficient $\eta$ is a tunable hyperparameter and $\circ$ denotes function composition. In the regularized model R-$\mathcal{M}$, we do not make any other changes to the based model $\mathcal{M}$ apart from using the new loss function $\mathcal{L}_{\text{R-}\mathcal{M}}$ during training. A schematic illustration is shown in Fig. 3 (a).

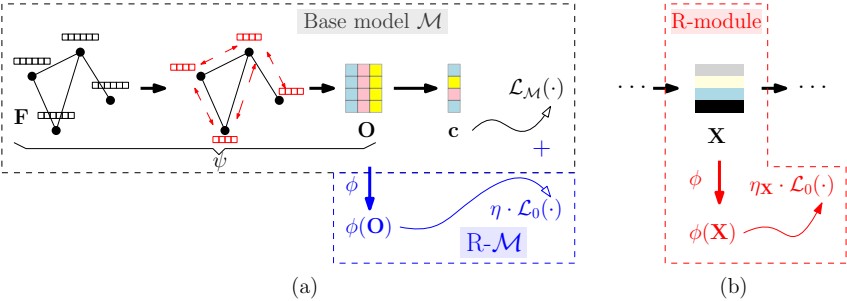

Figure 3: This figure illustrates in (a) how R-$\mathcal{M}$ is constructed upon the base model $\mathcal{M}$, and in (b) an R-module that can be used at different places in GNN models.

### 3.4 FURTHER DISCUSSIONS

Most regularization methods introduce a penalty to supplement the base model loss. For example, in the recent work, P-reg (Yang et al., 2021) proposes to apply a propagation matrix (e.g., the normalized adjacency matrix) to the output features. The model penalizes large discrepancies, measured by a metric such as squared error distance, between the original and transformed features. In LEreg (Ma et al., 2021), intra-energy and inter-energy losses are introduced. Both are variants of the total variation (1). The novelty lies in introducing a "merged graph" with each node representing a whole label class. All these works are related to the smoothness prior used in GSP theory (Section 2.1). In this paper, we take a fundamentally different view of graph signals by *treating a distribution as a signal*. This allows a *principled* and *straightforward* adaption of existing GSP approaches, as long as we have suitable notions of the signal's total variation. Moreover, non-uniformity does not have a counterpart for ordinary graph signals, which are essentially delta distributions.

Compared to GSP as well as other regularization methods such as P-reg and LEreg, a salient feature of our method is that the matrix $\boldsymbol{I}_n - \boldsymbol{A}_{\mathcal{G}}$ in $\mathcal{L}_0$ is in general not positive semi-definite. Therefore, the infimum of $\mathcal{L}_0(\boldsymbol{X})$ can be $-\infty$ if the domain of the entries of $\boldsymbol{X}$ is unbounded. This is another reason why restricting to distributional graph signals is essential.

The proposed regularization is primarily for node classification as distributional graph signals can be interpreted as the likelihoods of class labels. However, the method can be extended to other tasks through an *R-module* (Fig. 3 (b)) that consists of the following steps:

- Apply $\phi$ (e.g., softmax) that turns a feature $\boldsymbol{X}$ into a matrix of probability weights $\phi(\boldsymbol{X})$.
- Plug $\phi(\boldsymbol{X})$ in the loss $\eta_{\boldsymbol{X}} \mathcal{L}_0$ with tunable coefficient $\eta_{\boldsymbol{X}}$.

Given a base model, multiple R-modules can be inserted at different places of the model pipeline, and all the losses are combined with the original loss of the model. The insight is that useful node features for graph learning tasks should be inherently associated with smooth and non-uniform distributional graph signals. We demonstrate this approach with link prediction and graph classification tasks in Appendix C.

# 4 EXPERIMENTS

In this section, we verify the empirical performance of our proposed regularization method based on both Euclidean and hyperbolic GNN models. We consider node classification under both transductive and inductive settings. The datasets used include Cora, Citeseer, Pubmed, (Amazon) Photo, CS, Airport, Disease, and PPI (Sen et al., 2008; Namata et al., 2012; Zhang & Chen, 2018; Shchur et al., 2018; Fey & Lenssen, 2019; Chami et al., 2019; Szklarczyk et al., 2016). Their statistics are given in Appendix A. For our regularization method, the source code is provided in Appendix B. No further tuning of the base model is needed. We compare with base models and benchmarks in Section 4.1, and with variants of our model in Section 4.2. Other graph learning tasks are discussed in Appendix C.

## 4.1 PERFORMANCE ON NODE CLASSIFICATION PROBLEMS

### 4.1.1 TRANSDUCTIVE LEARNING MODELS

In this subsection, we consider transductive tasks, in which there is a single graph containing both labeled training nodes and test nodes. The base models include GCN (Defferrard et al., 2016), GAT (Veličković et al., 2018), GraphSAGE (Hamilton et al., 2017) (abbreviated as SAGE) and GraphCON (Rusch et al., 2022) (abbreviated as CON). Implementation details are given in Appendix B.

In addition to comparison with base models, we use models having a similar structure to our approach as benchmarks. More specifically, we implement different versions of P-reg in (Yang et al., 2021), denoted by P-GCN and P-GAT (based on popular GNN models GCN, GAT), and different versions of LEReg (Ma et al., 2021), denoted by L-GCN and L-GAT. The parameters are tuned as suggested by the respective papers. We also have the Laplacian method, denoted by LAP, in which the final class label $c$ is used in the loss $\mathcal{L}_0$. For a fair comparison, tests are performed under the same hardware and software environment. Similar considerations are applied in Section 4.1.2 and 4.1.3.

We also compare with BVAT (Deng & Zhu, 2019), GAM (Stretcu et al., 2019), and GraphAT (Feng et al., 2021) (the results are taken from literatures though they are unreported for Photo and CS). The test accuracies (in %) are shown in Table 1. In each row, best performers are highlighted in blue and red for our approach and benchmarks, respectively. An underlined entry means no noticeable performance improvement over the base model is observed. In general, we see that our proposed regularized models improve upon their respective base models with significant performance gain in many cases (cf. Appendix E). Moreover, our method can match up with or even outperform many benchmarks.

Table 1: Transductive learning models.

| Dataset | Comparison w/ base models | | | | | | | | Benchmarks | | | | | | | |
|---|---|---|---|---|---|---|---|---|---|---|---|---|---|---|---|---|
| | GCN | R-GCN | GAT | R-GAT | SAGE | R-SAGE | CON | R-CON | LAP | P-GCN | P-GAT | L-GCN | L-GAT | BVAT | GAM | GraphAT |
| Cora | 81.0 ±1.07 | 83.4 ±0.24 | 83.1 ±0.61 | 83.7 ±0.40 | 81.8 ±0.65 | 82.6 ±0.31 | 83.9 ±1.12 | **84.4** ±1.52 | 80.7 ±0.89 | 80.4 ±0.61 | 81.2 ±0.65 | 81.5 ±0.45 | 82.6 ±0.75 | **83.4** ±1.01 | 82.3 ±0.48 | 82.5 ±0.60 |
| Citeseer | 70.6 ±0.48 | 73.6 ±0.70 | 68.7 ±0.69 | 70.7 ±1.23 | 70.6 ±0.47 | 72.0 ±0.28 | 73.5 ±1.27 | **74.6** ±1.16 | 70.7 ±0.64 | 70.4 ±0.87 | 70.3 ±0.76 | 70.7 ±0.59 | 69.0 ±0.94 | **73.9** ±0.54 | 72.7 ±0.62 | 73.5 ±0.38 |
| Pubmed | 79.2 ±0.29 | 79.9 ±0.40 | 76.9 ±0.77 | 76.9 ±0.77 | 77.8 ±0.31 | 78.7 ±0.53 | 79.1 ±1.16 | **80.0** ±1.18 | 79.1 ±0.73 | 79.0 ±0.31 | 76.9 ±0.95 | 78.7 ±0.25 | 73.5 ±0.47 | 78.0 ±0.84 | **79.6** ±0.63 | 79.1 ±0.20 |
| Photo | 91.1 ±0.55 | 91.4 ±0.49 | 91.4 ±0.41 | **92.5** ±0.42 | 90.0 ±0.36 | 92.2 ±0.42 | 90.3 ±0.64 | 91.1 ±0.43 | 91.1 ±0.43 | 91.6 ±0.20 | 91.6 ±1.10 | 91.5 ±0.28 | **92.0** ±0.45 | - | - | - |
| CS | 90.5 ±0.14 | 91.2 ±0.65 | 90.8 ±0.40 | 91.7 ±0.25 | 90.5 ±0.08 | **91.8** ±0.10 | 90.5 ±0.44 | 90.8 ±0.36 | 90.6 ±0.11 | 90.5 ±0.14 | 90.8 ±0.40 | 90.5 ±0.10 | **90.8** ±0.34 | - | - | - |

### 4.1.2 HYPERBOLIC MODELS

Base models considered in Section 4.1.1 generate embedding of nodes in Euclidean spaces. However, it is argued in works such as Chami et al. (2019) that for certain datasets such as Airport and Disease, one should consider embedding nodes in hyperbolic spaces and perform feature aggregation in the tangent spaces of hyperbolic spaces. Such a consideration is plausible as certain graphs are inherently hyperbolic (measured by $\delta$-hyperbolicity, see Bridson & Haefliger (1999)).

In this subsection for Airport and Disease datasets, we use hyperbolic versions of their Euclidean counterparts HGCN Chami et al. (2019), and HGAT Gulcehre et al. (2019) as base models. We also

consider the interactive model GIL that combines both Euclidean and hyperbolic approaches Zhu et al. (2020). The comparison results are shown in Table 2. Again, we see a general improvement by using the proposed regularization, which yields performance comparable with benchmarks.

Table 2: Hyperbolic models.

| Dataset | Comparison w/ base models | | | | | | Benchmarks | | | | |
|---|---|---|---|---|---|---|---|---|---|---|---|
| | HGCN | R-HGCN | HGAT | R-HGAT | GIL | R-GIL | LAP | P-HGCN | P-HGAT | L-HGCN | L-HGAT |
| Airport | 88.8 ±1.38 | 89.3 ±1.42 | 89.9 ±0.83 | 91.0 ±1.63 | 90.3 ±2.12 | **91.4** ±1.39 | 88.8 ±1.38 | 88.9 ±1.42 | **90.0** ±1.11 | 88.7 ±1.57 | 89.7 ±1.14 |
| Disease | 91.3 ±1.04 | **92.1** ±1.09 | 90.4 ±1.24 | 91.3 ±1.31 | 91.2 ±1.52 | 91.2 ±1.52 | 92.2 ±1.01 | 92.1 ±1.07 | 90.2 ±2.14 | **92.2** ±0.95 | 90.4 ±1.53 |

### 4.1.3 INDUCTIVE LEARNING MODELS

In contrast with transductive learning, inductive learning requires one to deal with unseen data outside the training set. For example, in the PPI dataset, different graphs correspond to different human tissues and we test two out of 24 graphs that are unseen during training. Though the citation datasets Cora, Citeseer, and Pubmed are for transductive learning, we modify the datasets following Mishra et al. (2021) that use the induced subgraph of training nodes during training. During validation and testing, we use induced subgraphs of validation nodes and testing nodes with training nodes, respectively. The nodes that are being predicted are unseen during training.

The base models are GCN, GAT, as well as GraphSAGE that is primarily designed for inductive learning tasks. The performance comparison between the base models and their regularized versions is shown in Table 3. The conclusion agrees with those observed in the previous subsections.

Table 3: Inductive learning models

| Dataset | Comparison w/ base models | | | | | | Benchmarks | | | | |
|---|---|---|---|---|---|---|---|---|---|---|---|
| | GCN | R-GCN | GAT | R-GAT | SAGE | R-SAGE | LAP | P-GCN | P-GAT | L-GCN | L-GAT |
| Cora | 72.7 ±1.35 | **73.2** ±1.20 | 70.6 ±1.89 | 72.0 ±1.77 | 72.8 ±0.77 | 73.1 ±1.13 | 72.8 ±1.22 | 73.4 ±0.75 | 72.8 ±1.34 | **73.6** ±1.15 | 72.3 ±0.80 |
| Citeseer | 65.5 ±1.33 | **66.4** ±1.27 | 63.4 ±2.57 | 63.8 ±2.18 | 64.6 ±1.31 | 65.0 ±1.15 | 66.1 ±0.98 | 65.7 ±1.92 | 63.0 ±1.24 | 65.9 ±2.55 | 63.5 ±1.80 |
| Pubmed | 73.2 ±1.57 | **73.8** ±0.95 | 72.8 ±1.01 | 73.8 ±1.15 | 73.3 ±0.90 | 73.5 ±0.81 | 73.5 ±1.52 | **73.5** ±1.19 | 73.4 ±0.85 | 73.4 ±1.49 | 72.9 ±1.42 |
| PPI | 70.2 ±2.06 | 71.3 ±0.51 | 97.6 ±0.54 | **98.2** ±0.11 | 72.6 ±1.37 | 72.6 ±1.37 | 70.8 ±0.70 | 69.7 ±2.04 | 98.0 ±0.23 | 69.8 ±1.42 | **98.2** ±0.12 |

### 4.2 ANALYSIS AND ABLATION STUDIES

We analyze our model with R-GCN, which has significant gain as compared with its base model (cf. Table 1). The graph $\mathcal{G}_0 = (\mathcal{V}_0, \mathcal{E}_0)$ is the main connected component of Cora. Specifically, we want to study whether R-GCN indeed generates distributional graph signals with desired properties. For smoothness (cf. Section 3.1), we have interpreted earlier $\phi(\boldsymbol{O})$ of the output features of $\boldsymbol{O}$ as weights of marginal distributions. For the column $\phi(\boldsymbol{O})_{:,1}$, we take the subvector indexed by $\mathcal{V}_0$, then normalize and denote it by $\boldsymbol{x}$ (analysis of other columns are in Appendix G). We compute its Fourier transform $\hat{\boldsymbol{x}}$ and inspect its high-frequency components. We show $\hat{\boldsymbol{x}}$ for different epochs in Fig. 4. We also show the spectral plots for the last epoch (epoch 200) of GCN and of the signal of (normalized) ground truth labels. We see a clear shrinkage of high-frequency components for R-GCN (epoch 200).

For non-uniformity (cf. Section 3.2), we collect in the set $\mathbb{K}_{\text{R-GCN}}$ (resp. $\mathbb{K}_{\text{GCN}}$) the probability weights for all the label classes and nodes, i.e., 18956 entries of $\phi(\boldsymbol{O})$ for epoch 200 of R-GCN (resp. GCN). Non-uniformity suggests that $\mathbb{K}_{\text{R-GCN}}$ contains less values near the average $1/7$ and more values near 1, as compared with $\mathbb{K}_{\text{GCN}}$. To verify, we compare $|\mathbb{K}_{\text{R-GCN}} \cap [1/7 - \epsilon_1, 1/7 + \epsilon_1]|$ with $|\mathbb{K}_{\text{GCN}} \cap [1/7 - \epsilon_1, 1/7 + \epsilon_1]|$, and $|\mathbb{K}_{\text{R-GCN}} \cap [1 - \epsilon_2, 1]|$ with $|\mathbb{K}_{\text{GCN}} \cap [1 - \epsilon_2, 1]|$. The plots for different choices of (small) $\epsilon_1, \epsilon_2$ are shown in Fig. 5. The results agree with our speculation.

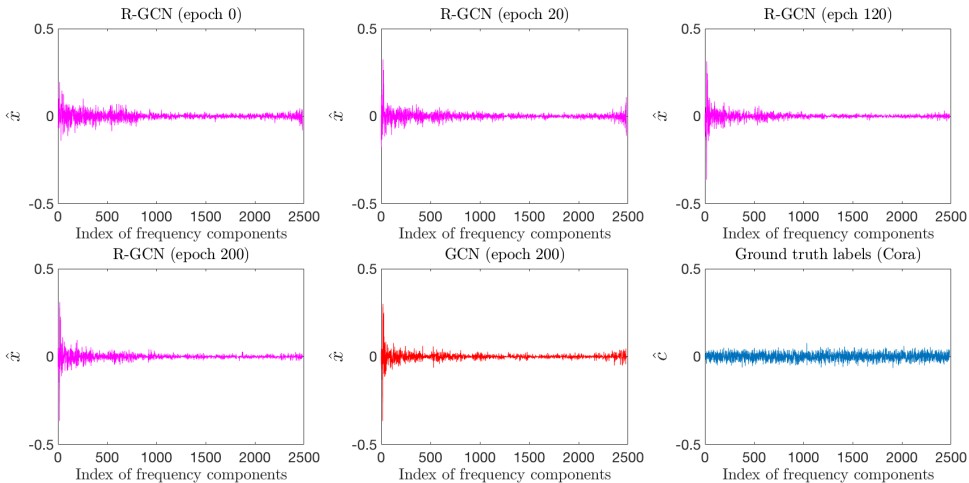

Figure 4: Spectral plots of (normalized) signals of probability weights and ground truth labels.

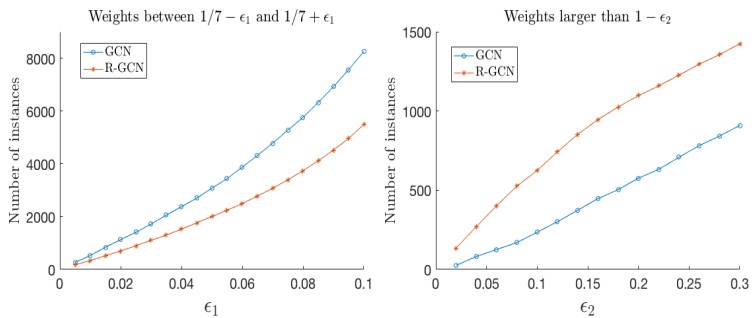

Figure 5: Number of instances of output probability weights within a given range.

Next, we conduct experiments on different R-GCN variants to validate the effectiveness of the components of our model. Recall that the newly introduced loss $\mathcal{L}_0$ has two components $\mathcal{L}_1$ and $\mathcal{L}_2$, accounting for smoothness and non-uniformity of distributional graph signals. We introduce modifications of R-GCN as follows: (a) R1-GCN: we only retain $\mathcal{L}_1$ in the loss; (b) R2-GCN: we only retain $\mathcal{L}_2$ in the loss, and (c) R3-GCN: we input $O$ directly in the loss without applying $\phi$. In this case, the entries of $O$ are not bounded, we only keep $\mathcal{L}_1$, which is positive semi-definite, in the loss (cf. Section 3.3). The results are shown in Table 4. We see that R-GCN remains the most effective as compared with its variants. This suggests that each component of R-GCN plays a useful role.

Table 4: Ablation study

|        | Cora | Citeseer | Pubmed |
|--------|------|----------|--------|
| R-GCN  | **83.4** $\pm$ 0.24 | **73.6** $\pm$ 0.70 | **79.9** $\pm$ 0.40 |
| R1-GCN | 80.8 $\pm$ 0.69 | 70.7 $\pm$ 0.67 | 79.2 $\pm$ 0.56 |
| R2-GCN | 82.0 $\pm$ 0.65 | 72.8 $\pm$ 0.92 | 79.5 $\pm$ 0.61 |
| R3-GCN | 69.6 $\pm$ 1.75 | 64.5 $\pm$ 1.79 | 75.7 $\pm$ 0.57 |

## 5 CONCLUSION

In this paper, we introduce the notion of distributional graph signals and total variations that measure the smoothness of such signals. Based on this and the concept of non-uniformity, we propose a regularization scheme that can be applied directly to enhance the performance of many GNN models. The method is analogous to the regularization method of a smooth signal prior in GSP.

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

## A    APPENDIX: DATA STATISTICS

In Table 5, we provide statistics of datasets used in Section 4.

Table 5: Dataset statistics

| Dataset | Nodes | Edges | Classes | Features |
|---------|-------|-------|---------|----------|
| Cora | 2708 | 5429 | 7 | 1433 |
| Citeseer | 3327 | 4732 | 6 | 3703 |
| Pubmed | 19717 | 44338 | 3 | 500 |
| Photo | 7487 | 119043 | 8 | 745 |
| CS | 18333 | 81894 | 15 | 6805 |
| Disease | 1044 | 1043 | 2 | 1000 |
| Airport | 3188 | 18631 | 4 | 4 |
| PPI | Ave. 2373 | Ave. 34171 | 121 | 50 |

## B    APPENDIX: MODEL IMPLEMENTATION

In this appendix, we provide the source code for the loss function $\mathcal{L}_0$ and the implementation of the regularization method. In addition, we give details of the base models used. For the source code, we assume PyTorch and the Deep Graph Library (dgl) are used.

In Fig. 6, we first import modules and packages (though not all packages are used for the other code segments below):

```python
import torch
import torch.nn as nn
import torch.nn.functional as F
import dgl
import dgl.nn as dglnn
from dgl.data import CoraGraphDataset, CiteseerGraphDataset, PubmedGraphDataset
from dgl import AddSelfLoop
import scipy
from scipy import sparse
from scipy.io import savemat
import argparse
import numpy as np
```

Figure 6: Import modules and packages.

In "my_loss" function (Fig. 7), we implement the loss $\mathcal{L}_0$ (cf. Section 3.3). For the inputs of "my_loss", "g" is a dgl graph, and "x" corresponds to $O$ in Section 3.3.

```python
def my_loss(g, x): #g is the graph and x is the output logits
    prob = F.softmax(x)   #Compute the marginal distributaions: "prob" contains probability weights
    num_nodes = g.number_of_nodes() #Find the number of nodes in the graph g
    adj = g.adjacency_matrix_scipy(return_edge_ids=False).astype(float) #The adjacency matrix of g
    laplacian = sparse.eye(num_nodes) - adj #Warning: not the true Laplacian of g but I_n-A_g in the paper
    laplacian = torch.from_numpy(laplacian.toarray()).float()
    y = torch.matmul(torch.matmul(torch.transpose(prob, 0, 1), laplacian), prob)
    y = torch.trace(y) #Compute the loss \mathcal{L}_0
    return y/num_nodes #Scale the result by number of nodes
```

Figure 7: "my_loss" function.

Suppose "loss1" (computed using "loss_fcn") is the loss of the base model, we compute "loss2" for $\mathcal{L}_0$ (Fig. 8). They are combined as in (6), using the tunable coefficient "eta".

```
eta = 0.2 #Tunable coefficient
loss1 = loss_fcn(logits[train_mask], labels[train_mask]) #The loss function of the base model
loss2 = my_loss(g, logits) #The loss \mathcal{L}_0 in the paper
loss = loss1+eta*loss2 #The loss of the regularized model
```

Figure 8: The regularization term.

As we have mentioned in the paper, we do not need to make any other changes to the based model. Details of the base models used in Section 4 are given as follows (links are for published works and not associated with the authors of the current paper):

- In Section 4.1.1, based models for GCN, GAT, SAGE and the datasets Cora, Citeseer, Pubmed are from `https://github.com/dmlc/dgl/tree/master/examples/pytorch`. For packages, we use torch 1.12.1 and dgl 0.9.0.

- In Section 4.1.1, based models for GCN, GAT, SAGE and the datasets Photo and CS are from `https://github.com/dmlc/dgl/tree/0.7.x/examples/pytorch`. For packages, we use torch 1.10.2, dgl 0.7.2 and cuda 10.2.

- In Section 4.1.1, the base model for CON is from `https://github.com/tk-rusch/GraphCON`. For packages, we use Torch-geometric 2.0.4, Torch-scatter 2.0.9, Torch-sparse 0.6.14, Torchdiffeq 0.2.3, and torch 1.10.1 py3.8 cuda11.3 cudnn8.2.0_0.

- In Section 4.1.2, the base models are from `https://github.com/CheriseZhu/GIL`. For packages, we use Torch 1.8.1+cu101, Torch-cluster 1.5.9, Torch-geometric 1.3.0, Torch-scatter 1.3.0, Torch-sparse 0.4.0 and Torch-spline-conv 1.2.1.

- In Section 4.1.3, the base models are from `https://github.com/dmlc/dgl/tree/0.7.x/examples/pytorch`. For packages, we use torch 1.10.2, dgl 0.7.2 and cuda 10.2.

Detailed model setups are contained in the respective github links. For example, according to `https://github.com/dmlc/dgl/blob/master/examples/pytorch/gcn/train.py`, for GCN and datasets Cora, Citeseer, Pubmed, two convolution layers with 16 hidden units are used. The dropout rate is set to be $0.5$. Adam optimizer is used with the learning rate $1e-2$ and weight decay $5e-4$. On the other hand, according to `https://github.com/dmlc/dgl/blob/master/examples/pytorch/gat/train.py`, for GAT and datasets Cora, Citeseer, Pubmed, two graph attention layers with 8 hidden units and 8 heads are used. The dropout rate is set to be $0.6$. Adam optimizer is used with a learning rate $5e-3$ and weight decay $5e-4$. As the regularization does not change the base model, the exact same setups are used.

In Table 6, we provide the values for the coefficient $\eta$ used in Section 4 (irrelevant fields are filled with "-"). We briefly describe the strategy of choosing $\eta$. We fix a lower bound ($= 10^{-5}$) and an upper bound ($= 1$) for $\eta$, both are loose. We perform a search analogous to binary search within the range based on validation performance. The scaling factor for the search can be different from 2: we use a large scaling factor to identify an interval with significant performance improvement and then perform a fine-scale search within the interval. If no performance improvement is observed within the initial range, then we declare the regularization does not show improvement for the given base model.

Table 6: Choices of $\eta$

|         | Cora | Citeseer | Pubmed | Photo | CS    | Airport | Disease | Cora (ind.) | Citeseer (ind.) | Pubmed (ind.) | PPI    |
|---------|------|----------|--------|-------|-------|---------|---------|-------------|-----------------|---------------|--------|
| R-GCN   | 0.2  | 0.3      | 0.02   | 0.005 | 0.05  | -       | -       | 0.001       | 0.0005          | 0.001         | 0.001  |
| R-GAT   | 0.25 | 0.25     | -      | 0.02  | 0.05  | -       | -       | 0.0005      | 0.01            | 0.05          | 0.0001 |
| R-SAGE  | 0.2  | 0.2      | 0.02   | 0.02  | 0.1   | -       | -       | 0.01        | 0.1             | 0.001         | -      |
| R-CON   | 0.01 | 0.05     | 0.001  | 0.01  | 0.005 | -       | -       | -           | -               | -             | -      |
| R-HGCN  | -    | -        | -      | -     | -     | 0.01    | 0.001   | -           | -               | -             | -      |
| R-HGAT  | -    | -        | -      | -     | -     | 0.01    | 0.001   | -           | -               | -             | -      |
| R-GIL   | -    | -        | -      | -     | -     | 0.01    | -       | -           | -               | -             | -      |

## C  Appendix: link prediction and graph classification

We follow Section 3.4 to apply the proposed regularization method to link prediction and graph classification.

For link prediction (Liben-Nowell & Kleinberg, 2007; Zhang & Chen, 2018), we want to predict whether two nodes in a network are likely to have a link. Suppose a base GNN model for link prediction is given. We insert an R-module (Fig. 3 (b)) to the last node feature matrix in the model pipeline. The loss of the R-module is added directly to the original loss. We test on Cora and Airport datasets with base models: GCN, GAT, HGCN, and GIL. The results are shown in Table 7. Except for HGCN, we see that the regularization can enhance the performance of the base models.

Table 7: Link prediction. The best performer is highlighted in blue.

|         | GCN | R-GCN | GAT | R-GAT | HGCN | R-HGCN | GIL | R-GIL |
|---------|-----|-------|-----|-------|------|--------|-----|-------|
| Cora    | 88.1 ±1.08 | 88.4 ±1.20 | 88.8 ±1.41 | 89.1 ±1.26 | 93.3 ±0.43 | $\underline{93.3}$ ±0.43 | 91.1 ±13.4 | **94.6** ±7.99 |
| Airport | 93.0 ±0.44 | 93.2 ±0.37 | 93.6 ±0.61 | $\underline{93.6}$ ±0.61 | 97.6 ±0.13 | $\underline{97.6}$ ±0.13 | 97.3 ±3.73 | **97.7** ±3.27 |

We next consider graph classification. For such a task, we want to determine the class label of each graph in a dataset containing multiple graphs. Graph classification is closely related to the theoretical problem of graph isomorphism test, and GIN (Xu et al., 2019) is a GNN model that explores such a connection. We use GIN and the variant GIN2 with the learnable importance of the target node compared to its neighbors, as base models. Similarly to link prediction, we insert an R-module (Fig. 3 (b)) to the last node feature matrix in the model pipeline. We test with bioinformatics datasets MUTAG and PTC (Yanardag & Vishwanathan, 2015), following the protocol described in (Xu et al., 2019) and report the 10-fold cross validation accuracy. Comparison results are shown in Table 8. The regularization indeed works for both models.

Table 8: Graph classification. The best performer is highlighted in blue.

|       | GIN | R-GIN | GIN2 | R-GIN2 |
|-------|-----|-------|------|--------|
| MUTAG | $87.7 \pm 8.80$ | $\mathbf{89.9} \pm 6.40$ | $86.7 \pm 6.77$ | $89.3 \pm 7.98$ |
| PTC   | $64.6 \pm 8.53$ | $64.8 \pm 5.91$ | $64.8 \pm 9.37$ | $\mathbf{66.3} \pm 9.63$ |

## D  Appendix: proofs of theoretical results

In this appendix, we discuss and prove a general result that implies Theorem 1. In addition, we also prove Lemma 1. We start with a computation of the Wasserstein distance.

Suppose $\mathbb{S} = \{s_1, \ldots, s_m\}$ is a finite discrete set and $d$ is the discrete metric on $\mathbb{S}$. For $\mu, \nu \in \mathcal{P}(\mathbb{S})$, let $(\mu(s_i))_{1 \leq i \leq n}$ and $(\nu(s_i))_{1 \leq i \leq n}$ be their respective probability weights.

**Lemma 2.**

$$W(\mu, \nu)^2 = \frac{1}{2} \sum_{1 \leq i \leq m} |\mu(s_i) - \nu(s_i)|.$$

*Proof.* Let $\gamma = \big(\gamma(s_i, s_j)\big)_{1 \le i, j \le m}$ be in $\Gamma(\mu, \nu)$. We have

$$\sum_{1 \le i \le m} \sum_{1 \le j \le m} \gamma(s_i, s_j) d(s_i, s_j)^2$$

$$= \sum_{1 \le i \le m} \sum_{1 \le j \ne i \le m} \gamma(s_i, s_j)$$

$$= \sum_{1 \le i \le m} \left( \sum_{1 \le j \le m} \gamma(s_i, s_j) - \gamma(s_i, s_i) \right) \tag{7}$$

$$= \sum_{1 \le i \le m} (\mu(s_i) - \gamma(s_i, s_i))$$

$$\ge \sum_{1 \le i \le m} (\mu(s_i) - \min(\mu(s_i), \nu(s_i)).$$

As $W(\mu, \nu)^2$ is defined by taking the infimum of the left-hand side over all $\gamma \in \Gamma(\mu, \nu)$, we have $W(\mu, \nu)^2 \ge \sum_{1 \le i \le m} \big(\mu(s_i) - \min(\mu(s_i), \nu(s_i))\big)$. By the same argument, we also have $W(\mu, \nu)^2 \ge \sum_{1 \le i \le m} \big(\nu(s_i) - \min(\mu(s_i), \nu(s_i))\big)$. Summing up these two inequalities, we have

$$2W(\mu, \nu)^2 \ge \sum_{1 \le i \le m} \big(\mu(s_i) + \nu(s_i) - 2\min(\mu(s_i), \nu(s_i))\big)$$

$$= \sum_{1 \le i \le m} |\mu(s_i) - \nu(s_i)|.$$

Therefore, to prove the lemma, it suffices to show that there is a $\gamma$ such that $\gamma(s_i, s_i) = \min\big(\mu(s_i), \nu(s_i)\big)$. For this, we prove a slightly more general claim: if non-negative numbers $(x_i)_{1 \le i \le m}$ and $(y_i)_{1 \le i \le m}$ satisfy $\sum_{1 \le i \le m} x_i = \sum_{1 \le j \le m} y_j = a$, then there are non-negative $(z_{i,j})_{1 \le i, j \le m}$ such that $\sum_{1 \le j \le m} z_{i,j} = x_i, 1 \le i \le m$, $\sum_{1 \le i \le m} z_{i,j} = y_j, 1 \le j \le m$, and $z_{i,i} = \min(x_i, y_i), 1 \le i \le m$.

We prove this by induction on $m$. The case for $m = 1$ is trivially true by taking $z_{1,1} = x_1 = y_1$. For $m \ge 2$, without loss of generality, we assume that $x_1 \ge y_1$ and $x_2 \le y_2$. Then we choose $z_{1,1} = y_1$, $z_{2,2} = x_2$, $z_{1,j} = 0, 1 < j \le m$ and $z_{i,2} = 0, 1 \le i \ne 2 \le m$. As a result, we form another two sequences of non-negative numbers $x_1 - y_1, x_3, \ldots, x_m$ and $y_2 - x_2, y_3, \ldots, y_m$ with both summing to $a - x_2 - y_1$. By the induction hypothesis, we are able to find non-negative $(z'_{i,j})_{1 \le i, j \le m-1}$ for the two new sequences of length $m - 1$ each. It suffices to let $z_{i,j} = z'_{i-1,j-1}$ for $i > 1$ or $j > 2$ and $z_{i,1} = z'_{i-1,1}$ for $i > 1$ (illustrated in Fig. 9). This proves the claim and hence the lemma. $\qquad \square$

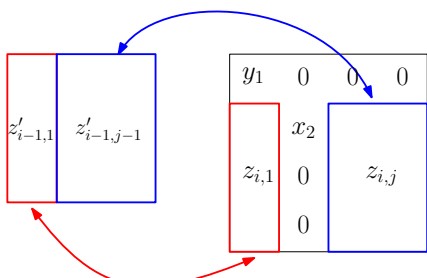

Figure 9: The relations between $z_{i,j}$ and $z'_{i,j}$.

To state and prove a general form of Theorem 1, we need to introduce a few more notions. We fix marginal distributions $\mathcal{N} = \{\mu_i : 1 \le i \le n\}$ with $\mu_i \in \mathcal{P}(\mathbb{S})$. For any pair of nodes $v_i$ and $v_j$ and $s \in \mathbb{S}$, define

$$\rho_{i,j}(s) = \mu_j(s)/\mu_i(s) \text{ if } \mu_j(s) \le \mu_i(s) \text{ and } 1 \text{ otherwise.} \tag{8}$$

More generally, if $P = (v_{i_0}, \ldots, v_{i_l})$ is a directed path on $\mathcal{G}$ from $v_{i_0}$ to $v_{i_l}$, then

$$\rho_P(s) = \prod_{0 \le j < l} \rho_{i_j, i_{j+1}}(s).$$

It is important to point out that $\rho_P$ can be computed directly as long as $\mathcal{N}$ is given.

In the graph $\mathcal{G}$, suppose $\mathcal{H}$ is a spanning tree and $v_0$ is a fixed (root) node. Let $\mathcal{E}_{\mathcal{H}}$ be the edge set of $\mathcal{H}$ and $\mathcal{E}' = \mathcal{E} \backslash \mathcal{E}_{\mathcal{H}}$. For each edge $e = (v_i, v_j)$, let $P_i$ (resp. $P_j$) be the unique path on $\mathcal{H}$ connecting $v_0$ and $v_i$ (resp. $v_j$). Moreover, $v_0$ is an endpoint of $P_i \cap P_j$, and let $v_k$ be the other endpoint of $P_i \cap P_j$. Denote by $Q_i$ (resp. $Q_j$) be the direct path (on $\mathcal{H}$) from $v_k$ to $v_i$ (resp. $v_j$) (see Fig. 10). We introduce

$$\mathfrak{t}_{i,j}(s) = \mu_i(s) + \mu_j(s) - 2\mu_k(s)\rho_{Q_i}(s)\rho_{Q_j}(s). \tag{9}$$

**Definition 5.** *Define*

$$\mathcal{T}_{\mathcal{H}, v_0}(\mathcal{N}) = \sum_{s \in \mathbb{S}} \sum_{(v_i, v_j) \in \mathcal{E}} \mathfrak{t}_{i,j}(s).$$

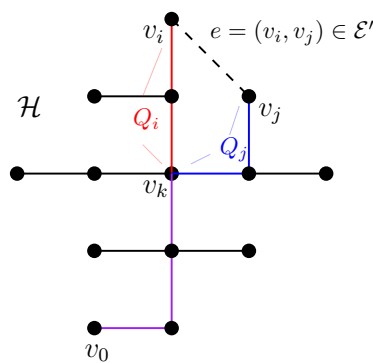

Figure 10: An example of paths $Q_i$ and $Q_j$.

We can compute the special case where $\mathcal{G} = \mathcal{H}$ is a tree. Notice that for an edge $e = (v_i, v_j) \in \mathcal{E}_{\mathcal{H}}$ directed from $v_i$ to $v_j$, then $v_k = v_i$ and $Q_i = \{v_i\}$, $Q_j = e$ and $\rho_{Q_i}(s) = 1$. If $\mu_i(s) \ge \mu_j(s)$, then $2\mu_k(s)\rho_{Q_i}(s)\rho_{Q_j}(s) = 2\mu_i(s) \cdot \mu_j(s)/\mu_i(s) = 2\mu_j(s)$. Hence, we have

$$\mu_i(s) + \mu_j(s) - 2\mu_k(s)\rho_{Q_i}(s)\rho_{Q_j}(s) = \mu_i(s) - \mu_j(s) = |\mu_i(s) - \mu_j(s)|.$$

The case $\mu_i(s) < \mu_j(s)$ is similar, and in summary

$$\mathfrak{t}_{i,j}(s) = \mu_i(s) + \mu_j(s) - 2\mu_k(s)\rho_{Q_i}(s)\rho_{Q_j}(s) = |\mu_i(s) - \mu_j(s)| \tag{10}$$

Therefore, for any $v_0$, we have

$$\begin{aligned}
\mathcal{T}_{\mathcal{H}, v_0}(\mathcal{N}) &= \sum_{s \in \mathbb{S}} \sum_{(v_i, v_j) \in \mathcal{E}} |\mu_i(s) - \mu_j(s)| \\
&= \sum_{s \in \mathbb{S}} \sum_{(v_i, v_j) \in \mathcal{E}} |\boldsymbol{\mu}_s(i) - \boldsymbol{\mu}_s(j)| = \mathcal{T}_1(\mathcal{N}),
\end{aligned} \tag{11}$$

where $\boldsymbol{\mu}_s(i)$ is same as $\mu_i(s)$ (cf. Section 3.1).

Following the notations in Section 3.1, we have the following generalization of Theorem 1.

**Theorem 2.** *For any spanning tree $\mathcal{H}$ of $\mathcal{G}$ and root node $v_0$, we have*

$$\mathcal{T}_2(\mathcal{N}) \le \mathcal{T}_1(\mathcal{N}) \le 2\mathcal{T}(\mathcal{N}) \le \mathcal{T}_{\mathcal{H}, v_0}(\mathcal{N}).$$

Before proving the result, we remark that although $\mathcal{T}_{\mathcal{H}, v_0}(\mathcal{N})$ is defined in a convoluted way, it can however be computed directly given $\mathcal{N}$, $\mathcal{H}$ and $v_0$. Therefore, the result gives computable upper and lower bounds of $\mathcal{T}(\mathcal{N})$.

*Proof.* As $|\boldsymbol{\mu}_s(i) - \boldsymbol{\mu}_s(j)| \leq 1$, it is trivially true that $\mathcal{T}_2(\mathcal{N}) \leq \mathcal{T}_1(\mathcal{N})$.

To show $\mathcal{T}_1(\mathcal{N}) \leq 2\mathcal{T}(\mathcal{N})$, we first claim that $\mathcal{T}(\mathcal{N}) = \inf_{\mu \in \Gamma(\mathcal{N})} \mathcal{T}(\mu)$ is achieved for some $\mu_0 \in \Gamma(\mathcal{N})$. The map $\alpha : \Gamma(\mathcal{N}) \to \mathbb{R}, \mu \mapsto \mathcal{T}(\mu)$ is continuous. On the other hand, $\Gamma(\mathcal{N})$ is a compact subset of a Euclidean space. This is because $\mathcal{P}(\mathbb{S}^n)$ is a bounded subset of $\mathbb{R}^{m^n}$ with the components corresponding to weights of the joint distribution. Moreover, $\Gamma(\mathcal{N})$ is closed because the condition to have the prescribed marginal distributions is a set of linear conditions. By the extreme value theorem, $\inf \alpha$ is achieved for some $\mu_0 \in \Gamma(\mathcal{N})$. For each edge $(v_i, v_j) \in \mathcal{E}$, let $\mu_{0,i,j}$ be the marginal distribution of the pair $(v_i, v_j)$. The marginals of $\mu_{0,i,j}$ are $\mu_i$ and $\mu_j$ at $v_i$ and $v_j$ respectively. We have

$$
\begin{aligned}
\mathcal{T}(\mathcal{N}) = \mathcal{T}(\mu_0) &= \int \sum_{(v_i,v_j)\in\mathcal{E}} d(\boldsymbol{x}(i), \boldsymbol{x}(j))^2 \, \mathrm{d}\mu_0(\boldsymbol{x}) \\
&= \sum_{(v_i,v_j)\in\mathcal{E}} \int d(\boldsymbol{x}(i), \boldsymbol{x}(j))^2 \, \mathrm{d}\mu_0(\boldsymbol{x}) \\
&= \sum_{(v_i,v_j)\in\mathcal{E}} \int d(\boldsymbol{y}(i), \boldsymbol{y}(j))^2 \, \mathrm{d}\mu_{0,i,j}(\boldsymbol{y}) \\
&\overset{\text{Def. 2}}{\geq} \sum_{(v_i,v_j)\in\mathcal{E}} W(\mu_i, \mu_j)^2 \\
&\overset{\text{Lem. 2}}{=} \frac{1}{2} \sum_{(v_i,v_j)\in\mathcal{E}} \sum_{s\in\mathbb{S}} |\boldsymbol{\mu}_s(i) - \boldsymbol{\mu}_s(j)| = \frac{1}{2}\mathcal{T}_1(\mathcal{N}).
\end{aligned}
$$

We now prove $2\mathcal{T}(\mathcal{N}) \leq \mathcal{T}_{\mathcal{H},v_0}(\mathcal{N})$. Given a spanning tree $\mathcal{H}$ and node $v_0$, we construct $\mu_{\mathcal{H},v_0} \in \Gamma(\mathcal{N})$ using ideas from the theory of Bayesian networks (Bishop, 2006) as follows. We make $\mathcal{H}$ directed by requiring that each edge is pointed away from the (root) node $v_0$. As a consequence, each node has at most 1 incoming edge. Let $\boldsymbol{x} = (\boldsymbol{x}_i)_{1 \leq i \leq n}$ be the random vector with $\boldsymbol{x}_i$ the (random) label at $v_i$. For each directed edge $(v_i, v_j)$ in $\mathcal{H}$, let $\bar{\mu}_{i,j} \in \Gamma(\mu_i, \mu_j)$ be a distribution that realizes $W(\mu_i, \mu_j)$, which give a conditional probability weights

$$
\boldsymbol{p}_{i,j} = \{p_{i,j}(s, s') : s, s' \in \mathbb{S}\} = \{p(\boldsymbol{x}_j = s' \mid \boldsymbol{x}_i = s) : s, s' \in \mathbb{S}\}.
$$

By Bishop (2006) Section 8.1 (8.5), there is a distribution $\mu_{\mathcal{H},v_0} \in \Gamma(\mathcal{N})$ such that its marginal for each edge $(v_i, v_j) \in \mathcal{E}_{\mathcal{H}}$ is $\mu_{i,j}$. For each edge $(v_i, v_j) \in \mathcal{E}'$, let $\mu'_{i,j}$ be the marginal of $\mu_{\mathcal{H},v_0}$ to the pair $(v_i, v_j)$. As $\mu_{i,j}$ realizes $W(\mu_i, \mu_j)$, by Lemma 2, $\mu_{i,j}(s, s) = \min(\mu_i(s), \mu_j(s))$ for each $s \in \mathbb{S}$. In particular, $p_{i,j}(s, s) = \rho_{i,j}(s)$ (cf. (8)). More generally, if $P$ is a directed path in $\mathcal{H}$ from $v_i$ to $v_j$, then the following inequality holds

$$
p_{i,j}(s, s) \geq \rho_P(s). \tag{12}
$$

According to the definition, we have $\mathcal{T}(\mathcal{N}) \leq \mathcal{T}(\mu_{\mathcal{H},v_0}) = \mathcal{S}_{\mathcal{E}_{\mathcal{H}}} + \mathcal{S}_{\mathcal{E}'}$. The summand

$$
\mathcal{S}_{\mathcal{E}_{\mathcal{H}}} = \sum_{(v_i,v_j)\in\mathcal{E}_{\mathcal{H}}} \int d(\boldsymbol{y}(i), \boldsymbol{y}(j))^2 \, \mathrm{d}\mu_{i,j}(\boldsymbol{y})
$$

is the summation over the edges of $\mathcal{E}_{\mathcal{H}}$, while

$$
\mathcal{S}_{\mathcal{E}'} = \sum_{(v_i,v_j)\in\mathcal{E}'} \int d(\boldsymbol{y}(i), \boldsymbol{y}(j))^2 \, \mathrm{d}\mu'_{i,j}(\boldsymbol{y})
$$

is the summand over $\mathcal{E}'$. For $\mathcal{S}_{\mathcal{E}_{\mathcal{H}}}$, we have seen that for each edge $(v_i, v_j) \in \mathcal{E}_{\mathcal{H}}$

$$
\begin{aligned}
&\int d(\boldsymbol{y}(i), \boldsymbol{y}(j))^2 \, \mathrm{d}\mu_{i,j}(\boldsymbol{y}) \\
&\overset{\text{Lem. 2}}{=} \frac{1}{2} \sum_{s\in\mathbb{S}} |\mu_i(s) - \mu_j(s)| \\
&\overset{(10)}{=} \frac{1}{2} \sum_{s\in\mathbb{S}} \big(\mu_i(s) + \mu_j(s) - 2\mu_k(s)\rho_{Q_i}(s)\rho_{Q_j}(s)\big),
\end{aligned}
$$

where the right-hand-side is the term $\frac{1}{2}\mathsf{t}_{i,j}(s)$ (cf. (9)) that corresponds to $(v_i, v_j)$ in $\frac{1}{2}\mathcal{T}_{\mathcal{H},v_0}(\mathcal{N})$.
Consider $(v_i, v_j) \in \mathcal{E}'$ and we first notice the following identity:

$$2 \int d(\boldsymbol{y}(i), \boldsymbol{y}(j))^2 \, \mathrm{d}\mu'_{i,j}(\boldsymbol{y}) \stackrel{(7)}{=} \sum_{s \in \mathbb{S}} \mu_i(s) + \mu_j(s) - 2\mu'_{i,j}(s, s).$$

To show the summand is bounded by $\mathsf{t}_{i,j}(s)$, it suffices to show $\mu'_{i,j}(s, s)$ is bounded below by $\rho_{Q_i}(S)\rho_{Q_j}(s)\mu_k(s)$, with $v_k$ and paths $Q_i, Q_j$ be as in (9). We estimate using the construction of $\mu'_{i,j}$ based on the method of Bayesian network (on directed $\mathcal{H}$) as follows:

$$\begin{aligned}
\mu'_{i,j}(s, s) &\geq p(\boldsymbol{x}_i = s, \boldsymbol{x}_j = s, \boldsymbol{x}_k = s) \\
&= p(\boldsymbol{x}_i = s, \boldsymbol{x}_j = s \mid \boldsymbol{x}_k = s)\mu_k(s) \\
&= p_{k,i}(s, s)p_{k,j}(s, s)\mu_k(s) \\
&\geq \rho_{Q_i}(s)\rho_{Q_j}(s)\mu_k(s).
\end{aligned}$$

For the last equality, we use the fact that $Q_i \cap Q_j = \{v_k\}$; and hence $\boldsymbol{x}_i, \boldsymbol{x}_j$ are independent given $\boldsymbol{x}_k$ by the Bayesian network construction. The last line follows from (12). Consequently, the inequality that $2\mathcal{T}(\mathcal{N}) \leq \mathcal{T}_{\mathcal{H},v_0}(\mathcal{N})$ follows. $\qquad \square$

If we examine the formula of $\mathcal{T}_{\mathcal{H},v_0}(\mathcal{N})$ (by changing summation order), it can be decomposed into two parts $\sum_{(v_i,v_j) \in \mathcal{E}_{\mathcal{H}}}$ and $\sum_{(v_i,v_j) \in \mathcal{E}'}$. The former is essentially $\mathcal{T}_1$ of $\mathcal{N}$ on the tree $\mathcal{H}$. This is the reason that Theorem 2 implies Theorem 1. Formally we have the following.

*Proof of Theorem 1.* It suffices to prove the last statement for $\mathcal{G}$ being a tree. As we have seen in (11) that if $\mathcal{G}$ is a tree, then $\mathcal{T}_{\mathcal{H},v_0}(\mathcal{N}) = \mathcal{T}_1(\mathcal{N})$. Therefore, Theorem 2 implies $2\mathcal{T}(\mathcal{N}) = \mathcal{T}_1(\mathcal{N})$, and hence Theorem 1. $\qquad \square$

We end this section by proving Lemma 1.

*Proof of Lemma 1.* Using Lemma 2, we estimate

$$\begin{aligned}
&2 \sum_{1 \leq i \leq n} a_i W(\mu_{\boldsymbol{X},i}, \mathcal{U}(\mathbb{S}))^2 \\
&= \sum_{1 \leq i \leq n} a_i \sum_{1 \leq j \leq m} \left| \boldsymbol{X}_{i,j} - \frac{1}{|\mathbb{S}|} \right| \leq \sum_{1 \leq i \leq n} a_i \sum_{1 \leq j \leq m} (\boldsymbol{X}_{i,j} - \frac{1}{|\mathbb{S}|})^2 \\
&= \sum_{1 \leq i \leq n} a_i \sum_{1 \leq j \leq m} \boldsymbol{X}_{i,j}^2 - 2 \sum_{1 \leq i \leq n} a_i \sum_{1 \leq j \leq m} \frac{\boldsymbol{X}_{i,j}}{|\mathbb{S}|} + \sum_{1 \leq i \leq n} a_i \sum_{1 \leq j \leq m} \frac{1}{|\mathbb{S}|^2} \\
&= \mathrm{Tr}(\boldsymbol{X}^\top \boldsymbol{D} \boldsymbol{X}) - 2 \sum_{1 \leq i \leq n} a_i \frac{1}{|\mathbb{S}|} + \sum_{1 \leq i \leq n} a_i \frac{1}{|\mathbb{S}|} \\
&= \mathrm{Tr}(\boldsymbol{X}^\top \boldsymbol{D} \boldsymbol{X}) - \frac{1}{|\mathbb{S}|} \mathrm{Tr}(\boldsymbol{D}).
\end{aligned}$$

Therefore, $2 \sum_{1 \leq i \leq n} a_i W(\mu_{\boldsymbol{X},i}, \mathcal{U}(\mathbb{S}))^2 \leq \mathrm{Tr}(\boldsymbol{X}^\top \boldsymbol{D} \boldsymbol{X}) + C$ with $C = -\frac{1}{|\mathbb{S}|} \mathrm{Tr}(\boldsymbol{D})$, which is independent of $\boldsymbol{X}$.

The last statement of the lemma follows from the simple fact that for each $1 \leq i \leq n$, we have

$$\sum_{1 \leq j \leq m} \frac{1}{|\mathbb{S}|^2} \leq \sum_{1 \leq j \leq m} \boldsymbol{X}_{i,j}^2 \leq 1.$$

$\qquad \square$

# E APPENDIX: SIGNIFICANCE OF THE PERFORMANCE

In this appendix, we analyze the significance of the performance (in Section 4.1) of the proposed regularization method with the following setup. Suppose we have two models $A$ and $B$ with $A$ having higher average accuracy. To determine if the difference is significant, we perform the hypothesis test for the null hypothesis that the difference in accuracy between model $A$ and $B$ has $0$ mean. We compute the $p$-value. The smaller the $p$-value, the higher the confidence to reject the null hypothesis and to conclude that model $A$ has a better performance. A $p$-value less than $0.05$ is typically considered statistically significant.

In Section 4.1, we have $38$ comparisons between regularized models and based models. We perform the tests described in the previous paragraph and record the $p$-values. The boxplot of the $p$-values is shown in Fig. 11. It indicates that in most cases, our proposed regularization significantly improves the respective based models. On the other hand, there are $11$ comparisons between our approach and best benchmarks. We again perform the hypothesis test for each comparison and record the $p$-value. The boxplot of the $p$-values is shown in Fig. 11. We see that in many cases, our method significantly outperforms the benchmarks.

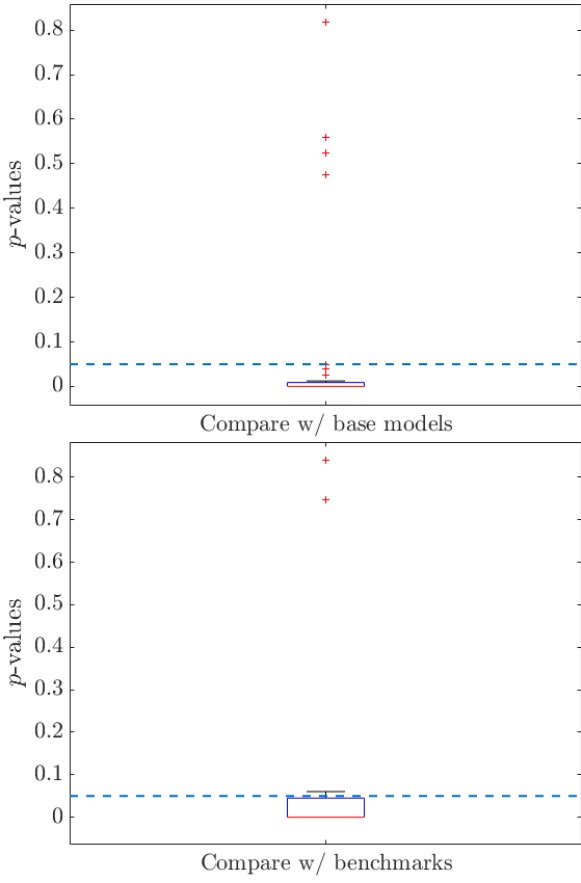

Figure 11: Boxplots of $p$-values. In each plot, the dashed line marks the $0.05$ threshold.

# F APPENDIX: MORE ON RELATED WORKS

Though it is argued in Yang et al. (2021) and supported by our evidence in Section 2.2 that Laplacian regularization may have its drawbacks, it has achieved a certain degree of success in earlier works Zhu et al. (2003); Zhou et al. (2004); Ando & Zhang (2007). These methods are based on the insight that neighboring nodes are likely to have the same labels. In the language of GSP, they intend to leverage the smoothness of the class label graph signal.

On the other hand, though graph signal smoothness has been fundamental in both GSP and GNNs, the negative effect of over-smoothing has also been examined (NT & Maehara, 2019; Oono & Suzuki, 2020)) and models are proposed to alleviate it. For example, apart from the more recent works such as Yang et al. (2021); Ma et al. (2021) that have already been discussed in detail, PairNorm proposed in Zhao & Akoglu (2020) encourages the similarity between connected nodes, and at the same time adds a negative term based on distances between disconnected pairs. MADReg in Chen et al. (2020) proposes to use step size limits to make the graph nodes receive less interference noise. In Feng et al. (2020), randomly dropping nodes is proposed to reduce the convergence speed of over-smoothing. Adding skip connections is also introduced in Li et al. (2019); Luan et al. (2019).

In our paper, we take a different point of view by not considering the smoothness of ordinary graph signals. Instead, we speculate that properties such as smoothness and non-uniformity of distributional graph signals may play important roles. Moreover, requiring a distributional graph signal to satisfy non-uniformity partially prevents the unfavorable situation that many connected nodes have similar marginal distributions that are approximately uniform. This can be viewed as a countermeasure to over-smoothing intrinsically contained in our approach.

## G  APPENDIX: ADDITIONAL PLOTS FOR MODEL ANALYSIS

We supplement Fig. 4 by showing spectral plots of signals of probability weights $\phi(\boldsymbol{O})_{:,i}, 2 \leq i \leq 7$ for the Cora dataset. The index $i$ corresponds to the $i$-th label class. From the plots, we observe that during the training of R-GCN, the high-frequency components indeed shrink for all the label classes. Compared with GCN, the last epoch of R-GCN has smaller high-frequency components for the 3rd and 5th label classes

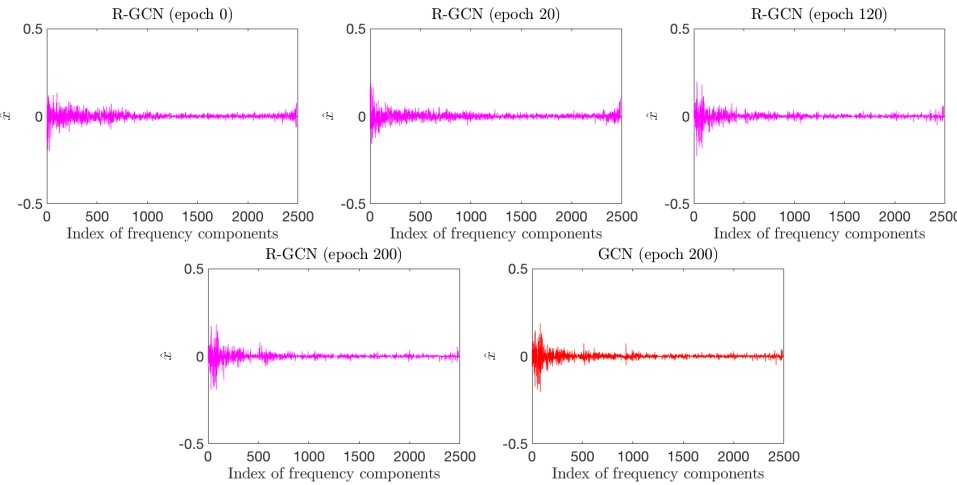

Figure 12: Spectral plots of (normalized) signals of probability weights for the 2nd label class.

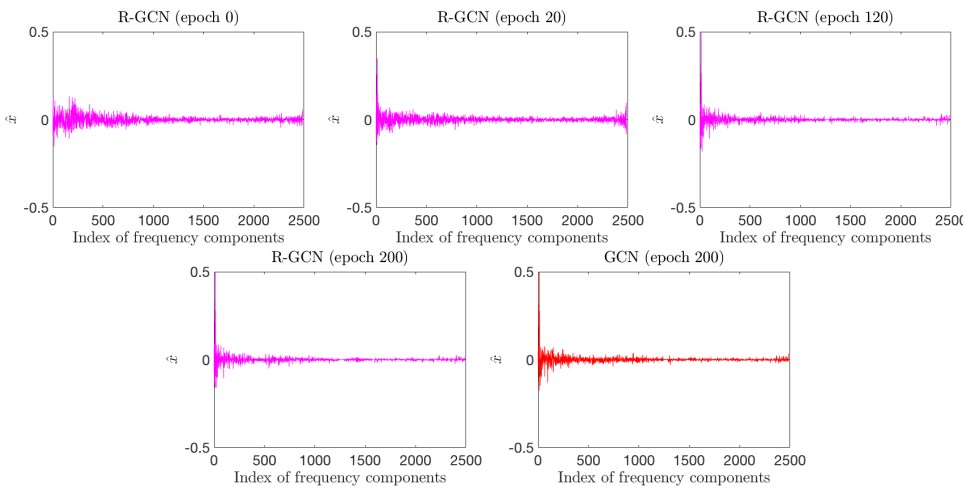

Figure 13: Spectral plots of (normalized) signals of probability weights for the 3rd label class.

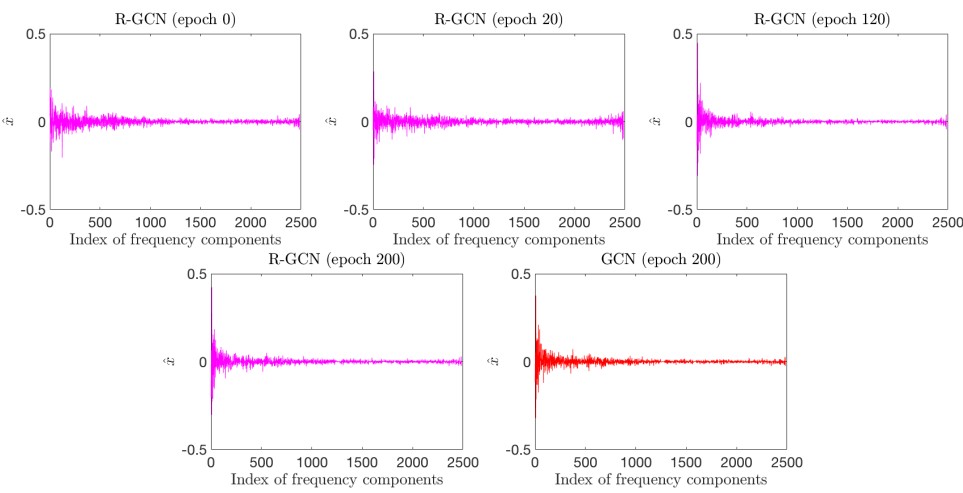

Figure 14: Spectral plots of (normalized) signals of probability weights for the 4th label class.

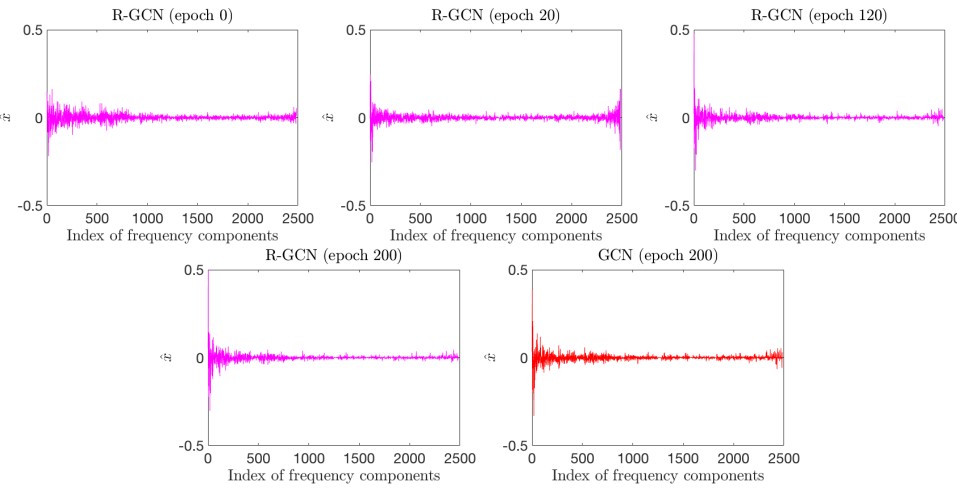

Figure 15: Spectral plots of (normalized) signals of probability weights for the 5th label class.

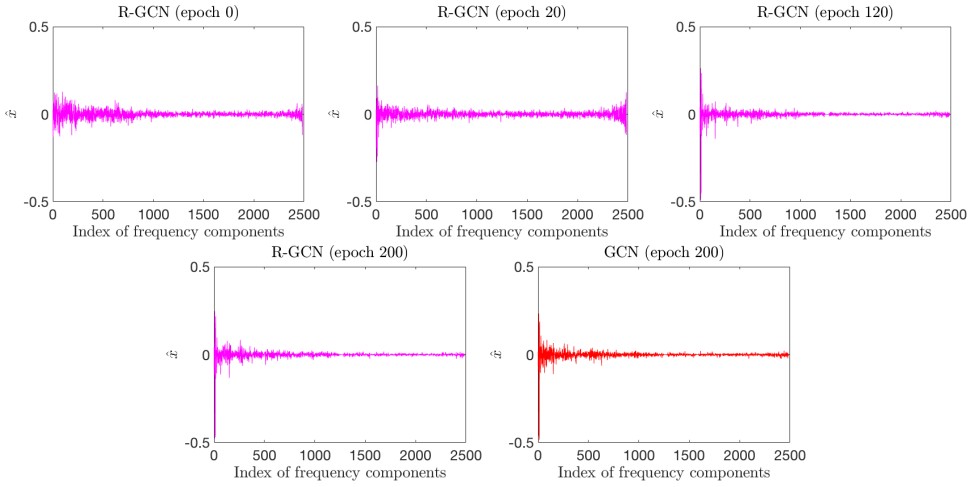

Figure 16: Spectral plots of (normalized) signals of probability weights for the 6th label class.

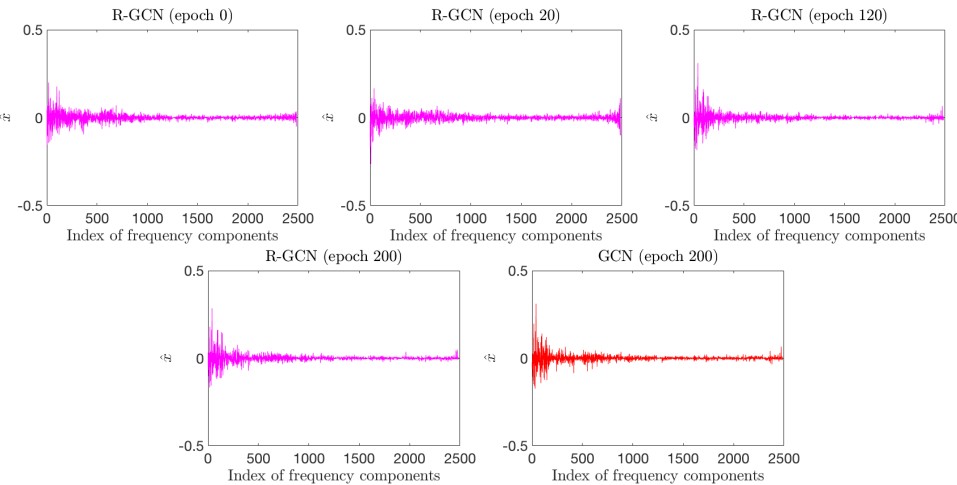

Figure 17: Spectral plots of (normalized) signals of probability weights for the 7th label class.

