# OpenReview forum: "Distributional Signals for Node Classification in Graph Neural Networks"
_ICLR.cc/2023/Conference — Submitted to ICLR 2023_

### Official Review · Reviewer_XoLJ · 2022-10-24

**Confidence:** 4
**Correctness:** 3
**Technical Novelty And Significance:** 3
**Empirical Novelty And Significance:** 3
**Recommendation:** 5

**Clarity, Quality, Novelty And Reproducibility:**

The methodology is clear and well written. The experimental design is lacking detail (in particular the details of the compared methods and the experimental settings).
Novelty: The work is novel and presents a very interesting approach for integrating graph signal processing techniques into graph neural networks.
Reproducibility: Code is not provided (only snippets are available). Based on the paper, it would be very challenging to reproduce the reported results.


**Details Of Ethics Concerns:**

Not applicable.

**Strength And Weaknesses:**

**Strengths**:
    1. This paper introduces the concept of *distributional graph signals*, generalizes important notions in GSP to the proposed frameworks, and constructs a novel graph regularization method, with a solid mathematical and theoretical foundation.
    2. The proposed regularization method shows empirical performance improvements compared to the selected base models and outperforms other selected regularization methods.

  - **Weaknesses**:
1. There is no comprehensive discussion on related work. The paper lacks a section devoted to related work, but beyond this, the embedded discussion (in the introduction and experiments sections, for example) does not provide an adequate introduction to compared methods and does not explain how the proposed approach differs and offers advantages. Example works on regularization that could be included in a discussion of related work are [R1-R4]; pioneering works on graph regularization include [R5-R6]. But beyond this, the paper needs to position the proposed technique within the general landscape of node classification methods.

 2. It appears that many of empirical improvements on the based model and outperformance over the benchmarks are not statistically significant. The paper mentions “noticeable” improvement, but there is no definition of this. There is no mention of testing to demonstrate a statistically significant improvement. In many cases the improvements appear to lie within 1 std, which doesn't augur well for the outcome of a carefully conducted test.

3.    The majority of the experiments are on Cora, Citeseer, Pubmed and PPI. These are small datasets that have received too much experimental attention – methods are over-tuned to a limited number of datasets. The limitations of the datasets do not allow one to properly examine the expressive capability of the models. One cannot trust that a technique will provide improvement for general graph datasets if experiments are only provided for these datasets. There are now many graph datasets to choose from and most recent GNN papers perform experiments on a much richer set.

4. The authors only verify the proposed enhancement method on the base models GCN, GAT and GraphSAGE (as well as GIN for graph-classification). All of these models were presented in papers published 4 or 5 years ago. The results would be much more persuasive if experiments were conducted with SOTA models as the base models.

[R1]: Lingxiao Zhao and Leman Akoglu. PairNorm: Tackling Oversmoothing in GNNs. In ICLR 2020.
[R2]: Deli Chen, Yankai Lin, Wei Li, Peng Li, Jie Zhou, and Xu Sun. Measuring and Relieving the Over-Smoothing Problem for Graph Neural Networks from the Topological View. In Proc. AAAI, 2020.
[R3]: Yang, Z., Cohen, W. and Salakhudinov, R., . Revisiting semi-supervised learning with graph embeddings. In Proc. ICML, 2016.
[R4]: Li, Q., Wu, X. M., Liu, H., Zhang, X., & Guan, Z. Label efficient semi-supervised learning via graph filtering. In *CVPR, 2019*.
[R5]: Zhu, Xiaojin, Zoubin Ghahramani, and John D. Lafferty. Semi-supervised learning using gaussian fields and harmonic functions. In Proc. ICML, 2003.
[R6]: Zhou, D., Bousquet, O., Lal, T., Weston, J., & Schölkopf, B. Learning with local and global consistency.  In Proc. NeurIPS, 2003.



**Summary Of The Paper:**

This paper addresses the limitation that the graph signal smoothing framework only works on continuous values, and proposes the concept of *distributional graph signals* in order to support discrete node labels.
The paper generalizes some important notions in graph signal processing (GSP), i.e., smoothness and non-uniformity, to distributional graph signals based on the Wasserstein metric.
The paper also proposes a general regularization method based on the proposed concept to enhance graph neural networks (GNNs) for the semi-supervised node classification task.


**Summary Of The Review:**

This paper provides an interesting and novel idea with good mathematical foundation. However, the work lacks a discussion of related work, and the experiments do not provide compelling evidence of the effectiveness of the proposed framework. The experiments need to be conducted more carefully with proper significance testing. Moreover, the experimentation should include more recent base models and richer datasets.

---

> ### Author Response · Authors · 2022-11-18
> **Response to the reviewer's comments regarding the weaknesses**
>
> Thank you for the helpful comments and suggestions. We have revised the paper to address the concerns.
>
> (1)  Regarding related works, due to space constraints, we were not able to include a discussion on all the related works in the main body of the paper, other than those most relevant works. Instead, in the revision, we follow Yang et al. (2021) and include a discussion on related works in Appendix F.
>
> (2) Regarding the significance of performance gain, we have performed statistical tests and detailed discussions are contained in the newly added ``Appendix E: Significance of the performance''. In summary, statistical tests suggest that in most cases, the proposed approach significantly improves upon based models and outperforms benchmarks.
>
> (3) Based on the suggestion, in the revision, we have performed additional experiments on (Amazon) Photo and CS datasets. Both datasets are larger and have much more edges as compared with older ones such as Cora, and Citeseer. The results are reported in Section 4.1.1 Table 1. The new experiments suggest that our approach improves upon base models and outperforms benchmarks for the new datasets.
>
> Regarding the diversity of datasets, in the updated version of the paper, there are $8$ different datasets studied in Section 4, and $2$ additional datasets are used in Appendix C for the graph classification task.
>
> (4) Based on the suggestion, in the revision, we have included the current state-of-the-art model GraphCON (abbreviated as CON) as a base model. The model is recently published in ICML 2022. The results are shown in Section 4.1.1 Table 1. We see an improvement in accuracy over the based models in all the cases.
>
> (5) As we are considering several different tasks, the setup, base models, and benchmarks are described in each subsection of Section 4.1 and Appendix C. Since all the experimental setups are standard, due to space constraints, we omit details by either giving brief descriptions or pointing out appropriate references. Fine experimental details depend on the base model used. There are altogether $38$ base models being used and it is maybe more practical to indicate how such information can be retrieved. Therefore, in Appendix B, we provide all the links for the base models. In the revision, we also provide details for certain models as illustrations of how one can obtain fine experimental details.
>
> Regarding codes, we have provided all the necessary information in Appendix B: Model implementation. The code segments provided are enough to reproduce the results. As mentioned in Appendix B, what a reader has to do is to take the code for a base model and insert the code segments provided at appropriate places (also indicated in Appendix B). We have also provided comments for the code segments so that readers can easily modify them if they use different python toolboxes.
>
> There is only one hyperparameter $\eta$ that requires tuning. In Appendix B Table 6, we have provided all the choices of $\eta$ in our experiments. Moreover, we have also outlined a strategy on how $\eta$ can be selected.
>
> In addition, we have also given detailed information on all the base models being used (by giving github links). Versions of python toolboxes used in our paper are also provided.
>
> We thank you once again for your review, which has definitely helped make our paper’s contributions stronger.

---

> ### Author Response · Authors · 2022-11-29
> **Regarding the paper revision**
>
> Thanks for your support! We deeply appreciate your comments that have been used to improve our paper revision. We would like to know whether our response addresses your concerns. We are also happy to answer if you have any further questions

---

### Official Review · Reviewer_34XZ · 2022-10-25

**Confidence:** 3
**Correctness:** 3
**Technical Novelty And Significance:** 3
**Empirical Novelty And Significance:** 3
**Recommendation:** 6

**Clarity, Quality, Novelty And Reproducibility:**

The paper is well written with source code also provided for reproducibility. The idea of applying GSP regularization at the output layer is nicely elaborated and can be easily adapted to different methods.

**Strength And Weaknesses:**

Pros:
- Provides a distributional view in the final softmax layer, which open possibility to add different GSP regularizations.
- Theoretical analysis on the bound of total variations/losses.
- No need to change the base model, and can apply to almost all existing framework.
- Works on both Euclidean space and Hyperbolic space models.
- Extensive empirical experiments with multiple baseline models in different real-world dataset.

Cons:
- Selection/Tuning of \ita is missing.
- Does the method works on different input feature space (Euclidean and Hyperbolic), with only one total variance (or the two surrogate T_1, T_2)? Same for transiting from transductive to inductive learning, why would the assumption still be true if the underlying signal are coming from different/unseen space?


**Summary Of The Paper:**

In this paper, the authors proposed a regularization term that can be added to traditional GNN training. The regularization term incorporate the defined distributional graph signals, which is a probability measure defined over the node label space, and shows significant improvement when compared to baseline models without regularization in empirical results.

**Summary Of The Review:**

Overall, I think the paper's idea is neat and empirical results show it work nicely with different input data in various learning task. Given the simplicity and versatility of method, I would recommend acceptance of the paper.

---

> ### Author Response · Authors · 2022-11-18
> **Response to the reviewer's questions.**
>
> Thank you for the comments and questions. We would like to provide information regarding how the issues raised are addressed in the revision.
>
> (a) Regarding the choice of $\eta$, in Appendix B: Model implementation, we have provided all the choices of $\eta$ in Table 6. In the revision, we have described our strategy of choosing $\eta$ in the last paragraph of Appendix B.
>
> (b) In principle, the method can also be applied to different input feature spaces, as long as: (1) the base model has a mechanism that combines features from different spaces into a single one, and (2) the test label set is the same as that in training. For example, the model GIL is a hybrid model that considers Euclidean and hyperbolic features simultaneously. In the last step of the model pipeline, features coming from Euclidean and hyperbolic subroutines are combined together. Therefore, our proposed regularization is applied at this stage of the base model. Moreover, if we regularize the output probability of the predicted label (as in most cases in the paper), this is less sensitive to the input feature space.
>
> The insight is that we can always have meaningful comparisons between distributions on the same sample space. This is guaranteed by conditions (1) and (2) in the previous paragraph. On the other hand, it would be theoretically unjustified if our regularization is applied to features of different natures directly.
>
> For the same reason, regarding inductive learning, it is implicitly assumed that the feature for each node is of the same nature or at least so when processed by a base model. Therefore, our method can be applied.
>
> (c) In the revision, we have conducted experiments on two more datasets (Amazon Photo and CS) and a more recent based model GraphCON. The results show the advantage of our approach. Moreover, we have included statistical significance tests of our results in Appendix E and a discussion on other related works in Appendix F.
>
> We thank you once again for your review, which has definitely helped make our paper’s contributions stronger.

---

> ### Author Response · Authors · 2022-11-29
> **Regarding our revision**
>
> Thanks for your support! We deeply appreciate your comments as well as those from other reviewers that have been used to improve our paper revision. We would like to know whether our response addresses all the concerns. We are also happy to answer if you have any further questions

---

### Official Review · Reviewer_Eo3M · 2022-10-25

**Confidence:** 2
**Correctness:** 3
**Technical Novelty And Significance:** 2
**Empirical Novelty And Significance:** 2
**Recommendation:** 5

**Clarity, Quality, Novelty And Reproducibility:**

The organization is somewhat clear but the arguments are not easy to follow. For example, it is unclear what Section 3.1 is trying to do. It felt like the paper is going to compute total variation for distributional graph signals, but it ends up simply defining l1 and l2 versions of standard graph regularization, only that it uses class distribution instead of node labels, but that is what is usually done anyway. So what exactly are we getting by defining regularization of distributional graph signals. The key difference seems to be the use of non-uniformity constraint, but the connection between Section 3.2 and 3.3 is not very clear. How does the definition in Section 3.3 follow from the arguments made in Section 3.2?

The paper does seem to have some original ideas, but the final proposed regularization does not seem that very different from standard regularizers.

**Strength And Weaknesses:**

# Strength:
1. The regularization of a GNN is an important problem.
2. The paper has some interesting ideas.
***
# Weaknesses:
1. While the organization of the paper is fairly clear. The arguments are not easy to follow.
2. The gains in the results are within a single standard deviation of the best baseline. It would be good to do a stat-sig test to see whether the observed gains are statistically significant or not.

**Summary Of The Paper:**

The paper proposes a new regularization for Graph Neural Networks. Signal smoothness in cases if discrete node classification task is hard to define. Instead, the paper proposes to look at the distribution of the class labels at per node level and define smoothness on the distributional graph signals. The proposed regularization additionally ensures that nodes do not end up having uniform probabilities. The paper then proposes to use the regularization at different layers of a GNN model so as to be able to extend it for tasks beyond node classification.

**Summary Of The Review:**

The paper proposes a regularization term for Graph Neural Networks. The paper proposes to define graph distributional signal and define a smoothness on this signal, however, the final proposed form does not seem to be very different from standard regularizers. Secton 3.1 does not seem to actually make any substantial point. Also, it is unclear how Section 3.3 follows from Section 3.2. The final proposed regularizer seems to have come from air, based on very thin justifications. The gains reported in the experiment section are well within a single standard deviation of the best baseline. So it is unclear whether the observed are statistically significant or not.

Based on these, I am inclined to mark this as marginally below acceptance.

---

> ### Author Response · Authors · 2022-11-18
> **Response to the reviewer's comments regarding significance of the results and clarity of the paper**
>
> Thank you for the helpful comments and suggestions.
>
> (a) We have revised parts of the paper based on the comments of all the reviewers to make the statements clearer and arguments easier to follow. Regarding the significance of performance gain, we have performed statistical tests and detailed discussions are contained in the newly added "Appendix E: Significance of the performance". In summary, statistical tests suggest that in most cases, the proposed approach significantly improves upon based models and outperforms benchmarks.
>
> (b) The principled definition of total variation $\mathcal{T}$ is given in Definition 3. It is a direct generalization of the Wasserstein metric, a widely used metric for probability spaces. Moreover, $\mathcal{T}$ also generalizes the total variation of traditional GSP, in which a traditional graph signal corresponds to a delta distribution in the proposed framework. However, as we point out in the paper, $\mathcal{T}$ is usually hard to calculate and we need an approximation that is easily computable. This is the reason that $\mathcal{T}_1$ and $\mathcal{T}_2$ are introduced. The relations among $\mathcal{T}, \mathcal{T}_1$ and $\mathcal{T}_2$ are given in Theorem 1. As we have discussed in the remark below Theorem 1, $\mathcal{T}_2$ will be used as a proxy of $\mathcal{T}$, as it can be computed readily. Moreover, requiring $\mathcal{T}$ to be small will force $\mathcal{T}_2$ to be small as well. This contributes to half of the final regularization term in Section 3.3.
> On the other hand, we propose in Section 3.2 to consider the non-uniformity of distributional graph signals. The non-uniformity is again in principle measured by the Wasserstein metric. An approximation is given by the term on the left-hand side of (5) in Lemma 1.
> In Section 3.3, we propose the final regularization term $\mathcal{L}_0$. It is a sum of two parts $\mathcal{L}_0 = \mathcal{L}_1 + \mathcal{L}_2$, in which $\mathcal{L}_1$ corresponds to $\mathcal{T}_2$ in Section 3.1 regarding smoothness and $\mathcal{L}_2$ is based on Lemma 1 leveraging non-uniformity.
> We hope the above explanation clarifies the relations between Section 3.1, Section 3.2, and Section 3.3. In the revision, we summarize the connections among these subsections in the first paragraph of Section 3 before Section 3.1.
>
> (c) In addition to the above changes, in the revision, we have conducted experiments on two more datasets (Amazon Photo and CS) and a more recent base model GraphCON. The results show the advantage of our approach. We also include a discussion on other related works in Appendix F.
>
> We thank you once again for your review, which has definitely helped make our paper’s contributions stronger.

---

> ### Author Response · Authors · 2022-11-29
> **Regarding the revision**
>
> Thanks for your support! We deeply appreciate your comments that have been used to improve our paper revision. We would like to know whether our response addresses your concerns. We are also happy to answer if you have any further questions

---

### Author Response · Authors · 2022-11-18
**Summary of major changes in the revision**

We would like to thank all the reviewers for their helpful comments. Based on them, we made the following major updates to the paper:

1. At the beginning of Section 3, we give an overview of the connections among Section 3.1-3.3. As Section 3.3 contains the regularization model, we indicate how Section 3.1 and Section 3.2 contribute to the final model.

2. In Section 4.1, we perform experiments on two new datasets bringing the total amount of datasets considered in the paper to 10.

3. In Section 4.1, we use a recent GNN model published in ICML 2022 as based model, suggested by one of the reviewers.

4. In Appendix A, statistics of the new datasets are given.

5. In Appendix B, we provide more information on experimental details. Moreover, we describe how the hyperparameter $\eta$ is tuned in additional to giving the values in different cases.

6. In the newly added Appendix E, we perform statistical tests to demonstrate the significance of our results.

7. In the newly added Appendix F, we include a discussion on related works and describe how our work may address certain issue in earlier works.

---

### Decision · Program_Chairs · 2023-01-20

**Decision:**

Reject

**Justification For Why Not Higher Score:**

Unfortunately, this paper fails to provide substantially novel insight. Indeed, the introduction of the distributional graph signal is not properly motivated because it does not lead to quite new and effective regularization. The numerical experiments are also not so much convincing.

**Justification For Why Not Lower Score:**

N/A

**Metareview: Summary, Strengths And Weaknesses:**

This paper introduces a notion of distributional graph signal, define its total variation norm and give characterization of different notions of norms such as $\mathcal{T}_1$ and $\mathcal{T}_2$. In addition to this, the authors propose to consider a sparsity inducing penalty for training. Then, the authors give a regularization term that combines the total variation related norm and the sparsity inducing penalty for learning graph signals. The proposed method is investigated through the numerical experiments.

In the literature of GNN research, it is an important research issue to explore an effective regularization to obtain better performance. This paper gives one different viewpoint on this issue.
On the other hand, its novelty is not really significant. Indeed, although they introduced the total variation norm on the distributional graph signal (which is one of the core contributions of this paper), the final proposal does not substantially make use of this notion. Actually, the total variation is approximated by $\mathcal{T}_2$ which is usual Laplacian based smoothing operator. In that sense, the contribution of this paper is not presented in a fully convincing way. Moreover, the experimental results do not show significant improvement of the proposed method.
In summary, we don't see much novel and informative insights from this paper. This paper requires substantial improvement by enhancing significance of the contribution.